# A Step Towards Worldwide Biodiversity Assessment: The BIOSCAN-1M Insect Dataset

**Zahra Gharaee**[3*], **ZeMing Gong**[4*], **Nicholas Pellegrino**[3*], **Iuliia Zarubiieva**[2,5],
**Joakim Bruslund Haurum**[7], **Scott C. Lowe**[5,8], **Jaclyn T.A. McKeown**[1,2], **Chris C.Y. Ho**[1,2],
**Joschka McLeod**[1,2], **Yi-Yun C Wei**[1,2], **Jireh Agda**[1,2], **Sujeevan Ratnasingham**[1,2],
**Dirk Steinke**[2†], **Angel X. Chang**[4,6†], **Graham W. Taylor**[2,5†], **Paul Fieguth**[3†]

[1]Centre for Biodiversity Genomics, [2]University of Guelph, [3]University of Waterloo,
[4]Simon Fraser University, [5]Vector Institute for AI, [6]Alberta Machine Intelligence Institute (Amii),
[7]Aalborg University and Pioneer Centre for AI, [8]Dalhousie University
`https://biodiversitygenomics.net/1M_insects/`

## Abstract

In an effort to catalog insect biodiversity, we propose a new large dataset of hand-labelled insect images, the BIOSCAN-1M Insect Dataset. Each record is taxonomically classified by an expert, and also has associated genetic information including raw nucleotide barcode sequences and assigned barcode index numbers, which are genetically-based proxies for species classification. This paper presents a curated million-image dataset, primarily to train computer-vision models capable of providing image-based taxonomic assessment, however, the dataset also presents compelling characteristics, the study of which would be of interest to the broader machine learning community. Driven by the biological nature inherent to the dataset, a characteristic long-tailed class-imbalance distribution is exhibited. Furthermore, taxonomic labelling is a hierarchical classification scheme, presenting a highly fine-grained classification problem at lower levels. Beyond spurring interest in biodiversity research within the machine learning community, progress on creating an image-based taxonomic classifier will also further the ultimate goal of all BIOSCAN research: to lay the foundation for a comprehensive survey of global biodiversity. This paper introduces the dataset and explores the classification task through the implementation and analysis of a baseline classifier. The code repository of the BIOSCAN-1M-Insect dataset is available at `https://github.com/zahrag/BIOSCAN-1M`

## 1 Introduction

Global change is restructuring ecosystems on a planetary scale, creating an increasingly urgent need to track impacts on biodiversity. Such tracking is exceptionally challenging because life is highly diverse: the biosphere comprises more than 10 million multicellular species [42]. Until recently, this complexity has meant that an Earth observation system for biodiversity was inconceivable, however the increased power of DNA sequencing and the recognition that living organisms can be discriminated by short stretches of DNA have revealed a way forward, which has become the central focus of the International Barcode of Life (iBOL) Consortium.

Discriminating organisms by DNA sequences [22, 6] can revolutionize our understanding of biodiversity, not only by providing a reliable species proxy for known and unknown species, but also by revealing their interactions and assessing their responses to changes in the ecosystem. This is essential to mitigate a looming mass extinction, where an *eighth of all species* may become extinct by 2100 unless there is a significant change in human behaviour [10, 11].

---

*Joint first author.

†Joint senior/last author.

37th Conference on Neural Information Processing Systems (NeurIPS 2023) Track on Datasets and Benchmarks.

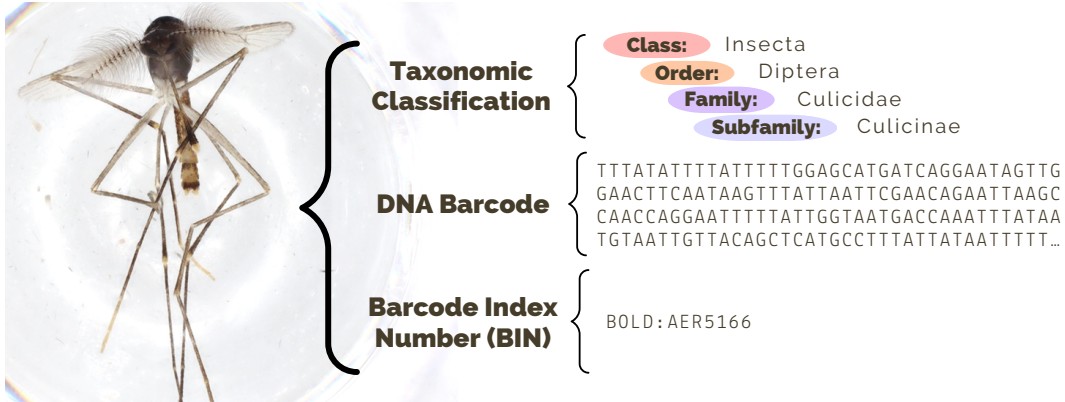

Figure 1: BIOSCAN-1M Insect dataset records contain high-quality microscope images of insects and labels including the taxonomic classification, raw DNA sequences, and Barcode Index Number (BIN). Pictured here is a mosquito of the subfamily *Culicinae*, the most populous subfamily of mosquitoes with species found around the world.

The BIOSCAN project [2], lead by iBOL, has the following three main goals:

1. Species discovery.

2. Studying the interactions between species.

3. Tracking and modelling species dynamics over geography and time.

To that end, BIOSCAN collects samples of multicellular life from around the world. Each sample is individually imaged, genetically sequenced and barcoded [22], and then classified by expert taxonomists. Of particular interest to the BIOSCAN project are *insects*, which constitute a great proportion of the Earth's species and many of which remain unknown. Indeed, it is estimated that 5.5 M insect species exist worldwide, of which only roughly one million have been identified [54, 23]. The rate of insect collection within the BIOSCAN project is increasing as the project progresses, such that 3 M insect specimens will be collected in 2023 and 10 M by 2028.

Using high-resolution photographs, human taxonomists can accurately classify insects from within their domain of expertise. However, human annotation cannot scale to the volume of samples needed to measure and track global biodiversity. Moreover, many taxonomists with highly specialized knowledge are leaving the practice and won't be replaced. Thus, the use of artificial intelligence and machine learning to process visual and textual information collected by the BIOSCAN project is crucial to the success of a planet-scale observation system. Classification of the insect images to their taxonomic group ranking is especially useful in regions of the world where the facilities required to perform genetic barcoding are not available. Indeed, even beyond this project, there are opportunities for computer vision to transform entomology [25].

This article has two main contributions.

1. The publication of the BIOSCAN-1M Insect image dataset, containing approximately 1.1 M high-quality microscope images, each of which is annotated by the insect's taxonomic ranking and accompanied by its raw DNA sequences and Barcode Index Number (BIN) [47], an example of which is shown in Figure 1.

2. The design and implementation of a deep model, classifying BIOSCAN-1M Insect images into specific taxonomic ranking groups, to serve as a baseline for future work utilizing this dataset.

## 2   Background and Related work

This section provides background on taxonomic classification, the use of genetic barcoding, and several challenges in the field of machine learning associated with our dataset.

## 2.1 Taxonomic Classification

In biology, taxonomic classification is the study of hierarchically categorizing lifeforms based on shared characteristics. In particular, Linnean taxonomy [7, 20, 32] forms the basis for the modern (generally accepted) system of taxonomy, of which the main hierarchical ranks are domain, kingdom, phylum, class, order, family, genus, and species, as shown in Figure 3. All insect life is part of the class *Insecta*.

Conventionally, expert taxonomists classify organisms based on their appearance and behaviour [7]. However, this approach is susceptible to both misclassification and lacks consensus throughout the community of taxonomists, since it is difficult to prove with certainty that a given classification is absolutely *correct*. This shortcoming of traditional taxonomy has prompted the use of classification heuristics, based on fairly concrete evidence in the form of genetic codes, that are sensitive to species identity.

Table 1 presents a comprehensive overview of both the number of unique categories and the degree to which the BIOSCAN-1M Insect dataset is labelled at each taxonomic rank. According to the table, a substantial number of samples lack labels at more specific taxonomic ranks. This limitation underscores the challenge of incomplete taxonomic labeling within the dataset. However, the ongoing efforts of expert curation are continuously improving this aspect, and future versions of the dataset are expected to include a more comprehensive taxonomic classification for a larger number of samples.

Based on the statistical analysis of the BIOSCAN-1M Insect dataset, it is evident that there are 3,441 distinct categories at the genus level and 8,355 at the species level. However, the amount of data to train such fine-grained classifiers is limited, with 254,096 samples classified by experts at the genus level and only 84,397 samples at the species level. This substantial class-number-to-data-size imbalance poses a notable challenge when training models for fine-grained classification, specifically at the genus and species level.

Table 1: An overview of the number of unique categories and number / proportion of expert-labelled samples within the BIOSCAN-1M Insect dataset at each taxonomic rank. Additionally, in the bottom row, the corresponding information is given in relation to Barcode Index Number (BIN), presented as a genetic alternative to taxonomic labels (species proxy). Observe that all samples have an associated BIN, and there are roughly $10\times$ more unique BINs than species labels.

| Taxonomic Level | Categories | Labelled Samples | Labelled (%) |
|---|---|---|---|
| Phylum | 1 | 1,128,313 | 100.0 |
| Class | 1 | 1,128,313 | 100.0 |
| Order | 16 | 1,128,313 | 100.0 |
| Family | 491 | 1,112,968 | 98.6 |
| Subfamily | 760 | 265,492 | 23.5 |
| Tribe | 535 | 60,477 | 5.4 |
| Genus | 3,441 | 254,096 | 22.5 |
| Species | 8,355 | 84,397 | 7.5 |
| **Barcode Index Number (BIN)** | 90,918 | 1,128,313 | 100.0 |

## 2.2 Genetic Barcoding and Barcode Index Numbers

DNA barcoding [22, 6] employs large-scale screening of one or a few reference genes for assigning unknown individuals to species, as well as aiding in the discovery of new species [43]. Barcoding is commonly used in several fields including taxonomy, ecology, conservation biology, diet analysis and food safety [49, 53]. It is faster and more accurate than traditional methods, which rely on the judgment of experts [46].

Barcoding is based on the use of a short, standardized segment of mitochondrial DNA, typically a portion of the *mitochondrial cytochrome c oxidase subunit I (COI) gene*, which is nearly always unique for different species. Once the DNA sequence is obtained, it can be compared to a reference library of known sequences to identify the species.

The concept of genetic barcoding can be taken a step further by mapping barcodes to clusters of organisms (characterized by their barcodes) with a *highly* similar genetic code, known as an

Operational Taxonomic Unit (OTU) [52, 5]. OTUs act as a proxy for species based on the high degree of genetic similarity exhibited by their members. To enable indexing, each OTU is assigned a Uniform Resource Identifier (URI), commonly referred to as the Barcode Index Number (BIN) [47], which offers a unique representation such that genetically identical taxa will be assigned the same BIN, and each BIN is registered in the Barcode Of Life Data system (BOLD) [1]. BINs additionally provide an alternative to the use of Linnean names, offering a genetics-based classification of organisms.

BOLD, the Barcode of Life Data System [1], is a pivotal resource in biodiversity science. It facilitates DNA barcode acquisition, storage, validation, and analysis, integrating molecular, morphological, and distributional data. BOLD hosts 17 million specimen records and 14 million barcodes, spanning over a million species worldwide. It plays a central role in species identification, genetic diversity exploration, and evolutionary studies. Launched in 2005, BOLD aims to identify all eukaryotic species and offers integrated analytical tools, comprehensive data management, and secure collaboration.

### 2.3 Machine Learning Challenges

As will be demonstrated in Section 3, the dataset exhibits two key characteristics corresponding to open problems in the field of machine learning.

**Class imbalance.** The degree to which the expected quantity of instances varies between classes is known as the class imbalance. In the context of a closed dataset, the class imbalance describes the disparity in number of examples among classes [27, 30]. As we describe in Section 3, and Figure 2 the published dataset exhibits a long-tailed class distribution whereby the distribution of class sizes closely follows a power-law, indicating that there is a substantial class imbalance. This represents a challenge due to the disproportionate amounts of available training data for majority vs. minority classes.

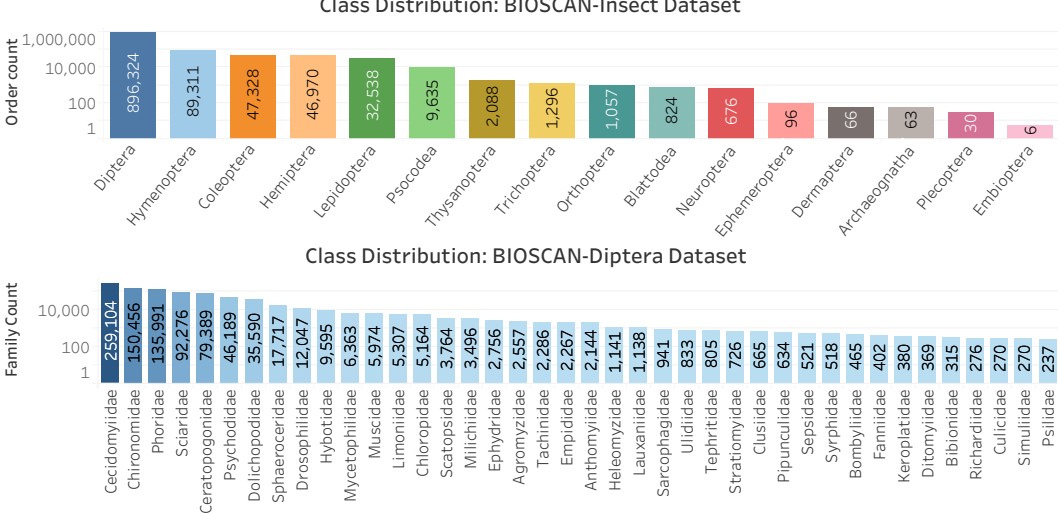

Figure 2: Class distribution and class imbalance in the BIOSCAN-1M Insect dataset. We focus on the 16 most densely populated orders (top) and the 40 most densely populated families (bottom).

**Hierarchical classification.** Classification problems involving data with labels that are inherently hierarchical present a unique challenge in comparison to simpler "flat" classification problems [50]. The outputs of hierarchical classification algorithms are defined over tree-like class taxonomies, where the relationship between parent and child nodes is given by the asymmetric "is-a" relationship. A basic example of this is the relationship that "all dogs are canines, but not all canines are dogs", whereby "dogs" would be a child node of the parent node "canines", which itself may be a child of "mammals". The dataset published here perfectly matches this paradigm and may be used to study novel approaches for handling the hierarchical classification problem. Note that the baselines we adopt in this paper do not pursue a hierarchical strategy but instead classify to fixed levels of the taxonomy: order and family. Hierarchical strategies are a topic of present and future work.

### 2.4 Biological Datasets

Image-based insect classification [39] most often finds use in agricultural settings, where Integrated Pest Management (IPM) systems are used to identify and count harmful insect pests [33, 51]. In combination with this, holistic systems capable of also identifying plant diseases through computer vision are a popular area of research [15, 12, 40].

Recently, DNA sequences have been analyzed [28] using tools from the field of Natural Language Processing [44], and in particular, through the application of bidirectional encoder representations from transformers (BERT) [14]. Indeed, BERT-based models have been used to taxonomically classify genetic sequences [24, 41]. Other recent work has used DNA barcodes as "side information" to perform zero-shot species-level recognition from images, albeit at a much smaller scale than the BIOSCAN-1M Insect dataset [4].

Perhaps the best known and largest biological dataset is iNaturalist [56, 26, 57], the 2021 version of which contains 2.7 M images from over 10,000 different species of plants, animals, and fungi, specifically with 2,526 species of insects with 663 k annotated insect images. Many insect-specific image datasets focus on insect as pests found in agricultural settings [62, 59, 58, 16, 63, 19, 37, 34]; the most prominent of which, the IP102 [62] dataset, contains roughly 75 k insect images, covering 102 species of common crop insect pests. 19 k of these are annotated with bounding boxes for object detection. In the space of plants, the PlantNet-300K [18] dataset has 306 k images labeled by species and was constructed by sampling the larger PlantNet database [3]. Table 2 highlights key biological datasets across a variety of domains and indicates the degree of class imbalance [13], $\beta$, which is defined as the ratio of the number of samples in the largest to the smallest class.

The BIOSCAN-Order and BIOSCAN-Diptera datasets, introduced in Section 4, refer to subsets of the full BIOSCAN-1M Insect dataset for use in order- and family-level classification, respectively. Observe that while there is significant variety in the imbalance factor among datasets, the imbalance of BIOSCAN-Order is *orders of magnitude* greater than that of all the other datasets. The Pl@ntNet-300K [18] dataset also has remarkably high class imbalance, exceeding that of the BIOSCAN-Diptera dataset. While a high imbalance ratio was expected of the BIOSCAN-1M Insect dataset based on the dataset's biological nature, the metric is *highly* sensitive due to its dependence on only the two most extreme classes.

The iNaturalist dataset encompasses a greater number of insect species than any pre-existing dataset. We measured the number of genera (plural of genus) and species that were common across both datasets. Of the 2,526 insect species in iNaturalist and 8,355 species annotated in BIOSCAN-1M Insect, only 153 genera and 62 species appeared in both datasets. This indicates the species in BIOSCAN-1M Insect predominantly do not appear in iNaturalist. Furthermore, based on the number of unique BINs present in the BIOSCAN-1M Insect dataset, it can be assumed that the dataset in fact encompasses almost 91 k distinct (possibly as yet unnamed) insect species, a far greater quantity than that of iNaturalist.

## 3 Dataset

This section describes the information made available through the publication of the BIOSCAN-1M Insect dataset, and details the procedures which generated the information.

### 3.1 BIOSCAN-1M Insect dataset resources

The BIOSCAN-1M Insect dataset provides four main sources of information about insect specimens. Each sample in the dataset consists of a biological taxonomic annotation, DNA barcode sequence, Barcode Index Number (BIN), and a RGB image of a single specimen. In the following sections, this information is described in detail.

#### 3.1.1 Biological taxonomy

The BIOSCAN-1M Insect dataset specifies biological taxonomic rank following the Linnean taxonomy as described in Section 2.1. In addition to the main groups shown in Figure 3, the dataset also provides the subfamily and subspecies ranks. The subfamily rank is an auxiliary (intermediate) taxonomic rank, the next below family but more inclusive than genus. Subspecies is a taxonomic rank below species, and it is used for populations that live in different areas and vary in size, shape, or other physical characteristics, but that can successfully interbreed. Finally, we also provide "Name"

Table 2: Summary of biological fine-grained and long-tailed datasets. Note that "iNaturalist-Insect" describes the subset of iNaturalist (2021 version) images that comprises insects. *For the BIOSCAN-1M Insect dataset, we report the number of unique Barcode Index Numbers (BINs) instead of the number of unique Linnean taxonomic species. The BIN is a mitochondrial DNA-based identifier which provides a species-like proxy of an organism and can be used as an alternative to Linnean taxonomy (see Section 2.2).

| Name / Citation | Domain | Images | Categories | Taxonomic Rank | Imbalance, $\beta$ |
|---|---|---|---|---|---|
| iNaturalist (2021) [57] | All | 2,686 k | 10,000 | Species | 1.97 |
| iNaturalist-Insect [57] | Insects | 663 k | 2,526 | Species | 1.97 |
| Pl@ntNet-300K [18] | Plants | 306 k | 1,000 | Species | 3,604.00 |
| Urban Trees [60] | Trees | 80 k | 18 | Species | 7.51 |
| IP102 [62] | Insects | 75 k | 102 | Species | 13.63 |
| NA Birds [55] | Birds | 48 k | 400 | Species | 15.00 |
| LeafSnap [31] | Plants | 31 k | 184 | Species | 8.00 |
| Pest24 [59] | Insects | 25 k | 24 | Common name / species | 493.95 |
| Flowers 102 [45] | Flowers | 8 k | 102 | Genus | 1.00 |
| **BIOSCAN-1M Insect** | Insects | 1,128 k | 90,918 | Barcode Index (BIN)* | 12,491.00 |
| **BIOSCAN-Order** | Insects | 1,128 k | 16 | Order | 156,856.75 |
| **BIOSCAN-Diptera** | Insects | 891 k | 40 | Family | 1,092.61 |

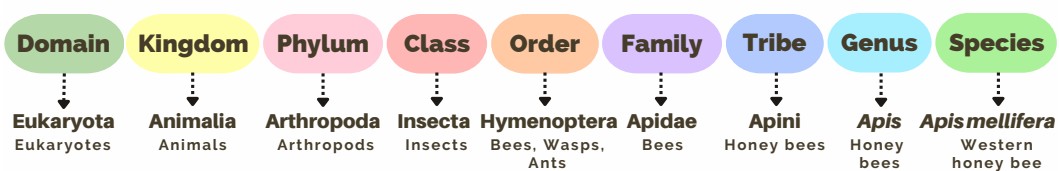

Figure 3: Biological taxonomic ranking and classification. Taxonomic ranks are shown in the top row, with the classification (i.e., labels) for the Western honey bee shown below.

to indicate the lowest (most specific) known rank label. For instance, if there exists a species name of an organism, "Name" will show that, but if the organism only has Family level ID, then "Name" will show that ID. Each sample has a name and there are in total 10,952 unique names labelled in the dataset.

Not all data samples have labels for all taxonomic ranks recognized in the BIOSCAN-1M Insect dataset. As an example, the family group of the BIOSCAN-1M Insect dataset is indexed by 494 distinct families, however, there are 16,067 data samples that are not associated with any of these families, since they were not classified by human taxonomists. As a consequence, there are many data samples that are not classified into finer-level groups like subfamily, tribe, genus, species, or subspecies. The lack of precise annotation at all ranks is one of the major challenges of the BIOSCAN-1M Insect dataset when performing classification tasks.

### 3.1.2 DNA Barcode and Indexing

Section 2.2 described the concept of genetic barcoding and the generation of Barcode Index Numbers (BINs). The BIOSCAN-1M Insect dataset contains genetic barcodes and BINs for all samples. This information is represented as the raw nucleotide barcode sequence, under the `Nuccraw` field, and the Barcode Index Number (BIN), denoted by `uri`. Independently, the field `processid` is a unique number assigned by BOLD to each record, and `sampleid` is an identifier given by the collector.

### 3.1.3 RGB images

The BIOSCAN-1M Insect dataset offers a wealth of information through its collection of insect images. The dataset contains high-resolution (2880×2160 pixel) RGB images in JPEG format; Figure 4 displays a selection of images representing insects from 16 most densely populated orders.

We have released multiple packages of the BIOSCAN-1M Insect dataset aimed at different purposes. These packages are organized into 113 chunks, each containing 10 k images. The packages include:

- Original JPEG Images stored in 113 zip files (2.3 TB).

- Cropped images organized into 113 zip files (151 GB).
- Resized original images which have a size of 256 px on their smaller side (26 GB).
- Resized cropped images having a size of 256 px on their smaller side (7 GB).

Additionally, we also provide the dataset in HDF5 archive format for both the resized original and cropped images.

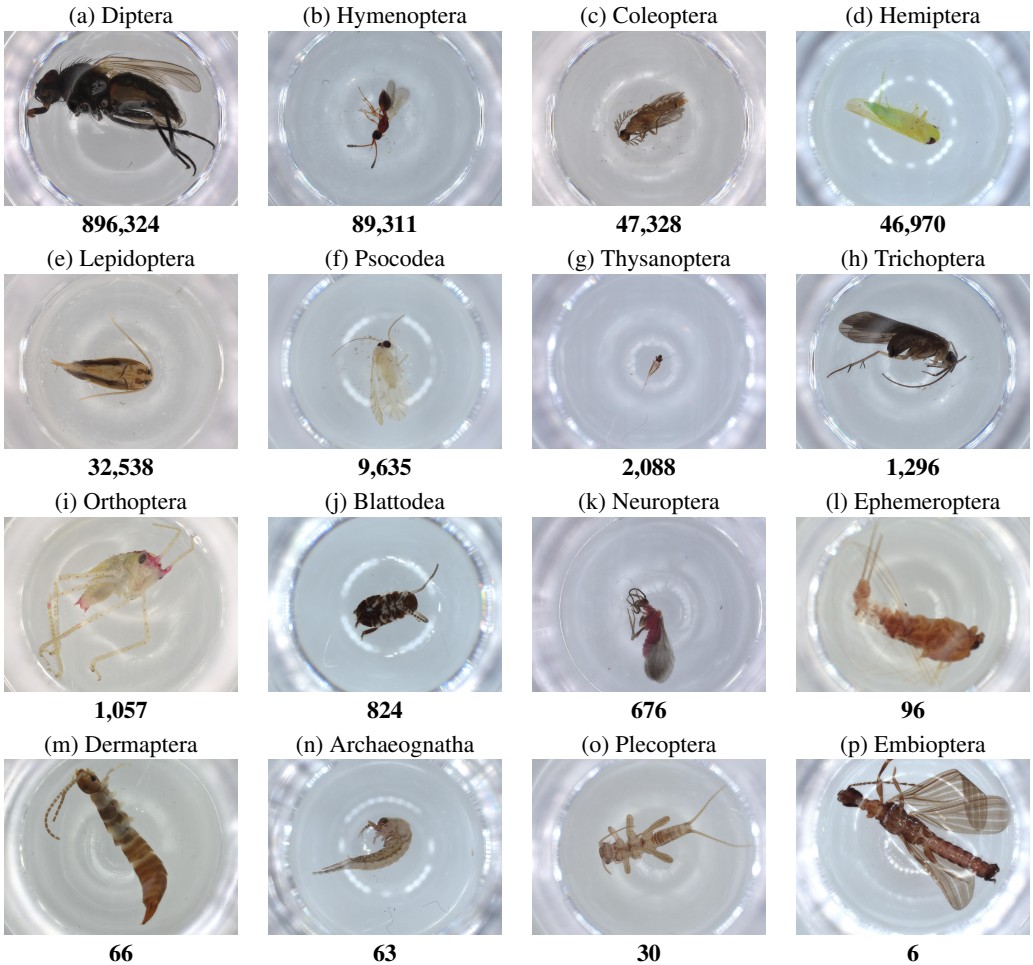

Figure 4: Examples of original insect images from 16 orders of the BIOSCAN-1M Insect dataset. The numbers below each image identify the number of images in each class, and clearly illustrate the degree of class imbalance in the BIOSCAN-1M Insect dataset. "Siphonaptera", "Strepsiptera" and "Zoraptera" are removed from classification experiments due to an insufficient number of samples.

## 3.2 BIOSCAN-1M Insect dataset generation

The BIOSCAN-1M Insect dataset consists of specimens mostly collected from three countries (Costa Rica, Canada, and South Africa) using Malaise traps. RGB images of the organisms were taken with a Keyence VHX-7000 microscope. Images are organized by workflow units: 96-well microplates of which 96 are used in a single sequencing run (9,120 samples at a time).

DNA barcodes of the organisms were generated using a high-throughput approach utilizing the Pacific Biosystems Sequel platform, which employs Single-molecule, real-time (SMRT) sequencing to generate long-read length DNA and cDNA. The taxonomic classifications were created by matching the generated barcodes to a reference library on the Barcode of Life Data System (BOLD) at the Centre for Biodiversity Genomics in Canada.

We provide a comprehensive metadata file alongside the RGB images, which includes taxonomic annotations, DNA barcode sequences, and data sample indexes and labels. The metadata file also contains image names and IDs to locate the corresponding images within the dataset packages. Additionally, it identifies the images associated with the training, validation, and test splits.

# 4 Experiments

By employing stratified class-based sampling, we methodically curated three subsets of varying sizes from the BIOSCAN-1M Insect dataset. Subsequently, we carried out two distinct sets of classification experiments, yielding a total of six datasets. The three subsets, namely Small, Medium, and Large, each comprising approximately 50 k, 200 k, and 1 M data samples, respectively. The initial set of experiments primarily revolves around classifying insect images into their 16 most densely populated taxonomic orders. Subsequently, the second set of experiments delves even deeper, focusing on classifying samples within the Order Diptera into their 40 most densely populated families.

## 4.1 Subset sampling and split mechanism

To create subsets of the BIOSCAN-1M Insect dataset, we followed a systematic two-step process. Initially, we sampled a subset exclusively from the Diptera Order, specifically selecting the 40 families with the highest number of members. This led to the creation of the BIOSCAN-Diptera dataset. Subsequently, we divided the BIOSCAN-Diptera dataset into separate train, validation, and test sets. Finally, we used these split sets of the BIOSCAN-Diptera dataset as a basis to construct the corresponding train, validation, and test sets for the BIOSCAN-Order dataset. Our methodology involved the use of stratified class-based sampling to preserve the class distribution consistently across all subsets, ensuring the integrity of our experiments.

Table 3: The total number of samples used in the six different sized subsets of the BIOSCAN-1M Insect dataset: The entries display the number of data samples in the train, validation, and test sets, as well as the number of classes for order-level (16 orders) and Order Diptera family-level (40 families) experiments.

| Dataset | Total | Train | Validation | Test | Categories |
|---|---|---|---|---|---|
| BIOSCAN-Order | 1,128,308 | 789,813 | 112,835 | 225,660 | 16 |
| BIOSCAN-Diptera | 891,338 | 623,937 | 89,135 | 178,266 | 40 |
| BIOSCAN-Order/Diptera Medium | 200,000 | 140,000 | 20,000 | 40,000 | 16/40 |
| BIOSCAN-Order/Diptera Small | 50,000 | 35,000 | 5000 | 10,000 | 16/40 |

The Small and Medium subsets are generated by sampling 50 k and 200 k data samples, respectively, from the train, validation, and test sets of the BIOSCAN-Order and BIOSCAN-Diptera datasets. In all of our classification experiments, we used class-based stratified sampling to split the dataset into train, validation and test sets. To this end, 70% of the samples of each class are randomly selected as training, 10% as validation, and 20% as test samples, as shown in Table 3.

The extreme class imbalances, which are an inherent characteristic of the BIOSCAN-1M Insect dataset, are addressed to some extent by having all classes represented in the train, validation and test sets. Classes with no samples for either split set are omitted. In the insect order-level classification (Figure 4), we have sufficient data samples for 16 out of 19 orders in the train, validation, and test sets. For the Diptera family-level classification, we focus on the 40 most populous families within Diptera.

## 4.2 Data preprocessing

To improve computational efficiency, we crop and resize the images to be 256 px on the smaller dimension. Preliminary experiments with ResNet-50 comparing original images with images that are cropped show that cropping can help model learning to converge more rapidly and lead to slightly better performance. Reducing the resolution to 256 px helps to reduce the size of the large dataset from 2.3 TB down to 26 GB for the original uncropped images, and from 151 GB down to 7 GB for cropped images. We choose to run experiments on the cropped and resized images due to the small size which allows for efficient data loading from disk.

The BIOSCAN-1M image datasets have insects with varying size, pose, color and shape. Due to these variations, cropping is not a simple task. We develop our cropping tool by fine-tuning a DETR [9]

model with ResNet-50 [21] backbone (pretrained on MSCOCO [35]) on a small set of 2,000 insect images annotated using the Toronto Annotation Tool Suite [29]. In DETR, the CNN-based feature extractor extracts a set of image features that are fed into a transformer-based encoder-detector. The detector takes a set of learned positional embeddings as object queries and uses them to attend to the encoder outputs. Each of the output decoder embeddings is then passed to a shared FFN which predicts whether there is an "insect" or "no object" and regresses the bounding box. The DETR model is further trained for 10 epochs with the AdamW optimizer with learning rate of 0.0001, weight decay of 0.0001 and a batch size of 8.

To crop the image, we apply our fine-tuned DETR model and take the predicted bounding box with the highest confidence score. The finalized cropping is determined as the predicted bounding box, extended equally in width and height by $0.4$ of the maximum dimension.

### 4.3 Classification model

To run classification experiments, we fine-tuned two different pre-trained models to extract deep visual features of insects from their RGB images. Our pre-trained models are ResNet-50 [21] and a transformer based model, ViT-Base-Patch16-224 [17]. During training, we take random 224×224 crops from the image as input, while during validation we take the center crop. To train our model, we used two loss functions, the Cross-Entropy (CE) as a baseline and the Focal loss, which is more suitable for datasets having class imbalances [36, 8, 13].

## 5 Results

The detailed hyperparameters used for our experiments are shown in Table 4. For conducting the experiments, we leveraged the computational resources provided by the Digital Research Alliance of Canada's Narval and Beluga clusters. To ensure efficient processing. Each experiment was performed using a single node equipped with 1 GPU, 10 CPUs per task, and a memory allocation of 128GB.

Table 4: Detailed hyperparameter settings of the experiments.

| Parameters | Settings | Parameters | Settings |
|---|---|---|---|
| Model | ResNet-50;ViT-B/16 | Batch-Size | 32 |
| Loss function | Cross-Entropy;Focal | Epoch | 100 |
| Optimizer | SGD | Num-Workers | 4 |
| Weight Decay ($\mu$) | 0.0001 | Image-Size (Train/Val) | 256 |
| Learning rate | 0.001 | Crop-Size (Train) | 224 |
| Momentum | 0.9 | Rand-Horizontal-Flip (Train) | Yes |
| K | [1, 3, 5, 10] | Centre-Crop (Val) | 224 |
| group-level | Order;Family | Dataset size | L/M/S |

We conducted a set of 24 trials, each executed with 3 distinct seeds. These trials were carried out to tackle classification tasks involving Insect-Order and Diptera-Family utilizing three dataset variations: Large, Medium, and Small. Our design encompassed the creation of 4 distinct models, integrating two distinct loss functions (Cross-Entropy and Focal) and two different pretrained backbones (ResNet-50 and ViT-B/16).

The combination of these diverse components led to the calculation of average performance across a range of seed values. Subsequently, the model that exhibited the highest average performance on the validation set was selected for further evaluation during inference, as depicted in Table 5. It's worth noting that models employing the ViT-B/16 backbone and Cross-Entropy loss function demonstrated superior performance across most of the experiments on the validation set, leading to their selection for inference using test data. For the Small and Medium datasets, the models underwent 100 epochs of training, while for the Large dataset, a lesser number of epochs were applied, as convergence was achieved on the validation set.

We evaluate the performance of our classification models using top-K accuracy, which extracts the K-predicted classes with the largest probabilities for each input sample and compares them with the ground-truth class label of the sample. If the ground-truth label is among the top-K predictions then the model counts it as a correct classification. The total counts are then divided by the total number of input samples to yield an average. We report test results of the best model from validation

performance for the micro, class-averaged macro top-K accuracy at $K \in [1, 3, 5]$ as well as micro and macro $F_1$ Scores Table 5.

Figure 5 shows the per-class top-1 test accuracy for the Order and Family classification of the Large dataset. Accuracy is quite high, above 90%, for most classes, decreasing mainly for classes with little training data.

Table 5: Top-K accuracy and class-averaged macro top-K accuracy based on the test sets of Insect-Order and Diptera-Family classification experiments using the Small, Medium and Large datasets.

| Classification | Dataset | Micro Top-K | | | Macro Top-K | | | F1-Scores | |
| | | Top-1 | Top-3 | Top-5 | Top-1 | Top-3 | Top-5 | Micro | Macro |
|---|---|---|---|---|---|---|---|---|---|
| Insect-Order | Small | 97.86 | 99.35 | 99.66 | 85.01 | 91.68 | 99.23 | 97.86 | 85.84 |
| | Medium | 99.14 | 99.77 | 99.88 | 85.58 | 97.68 | 98.22 | 99.14 | 87.36 |
| | Large | 99.69 | 99.96 | 99.98 | 90.61 | 98.14 | 99.32 | 99.62 | 92.65 |
| Diptera-Family | Small | 94.01 | 97.26 | 98.01 | 92.37 | 96.53 | 97.42 | 94.01 | 93.03 |
| | Medium | 96.66 | 98.34 | 98.77 | 91.81 | 96.37 | 97.20 | 96.66 | 92.77 |
| | Large | 97.59 | 98.85 | 99.23 | 91.20 | 95.86 | 96.72 | 97.59 | 91.45 |

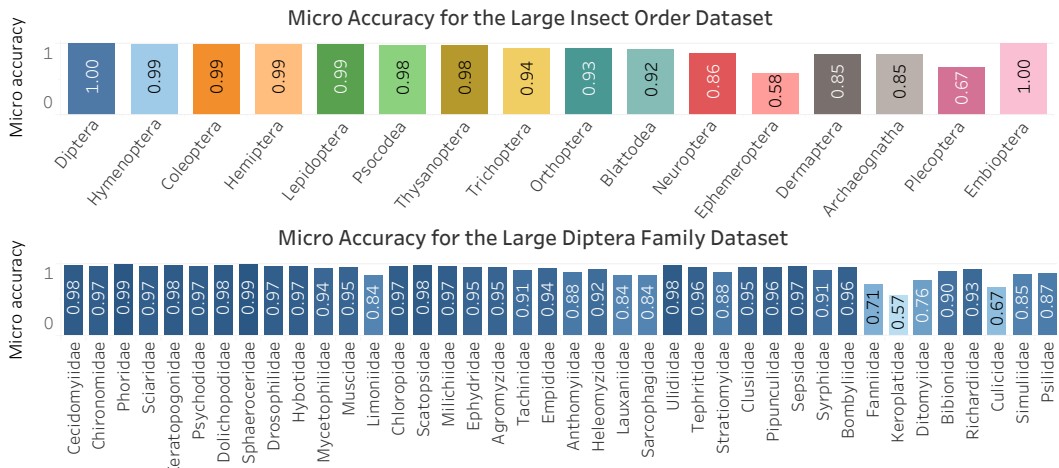

Figure 5: Per-class top-1 test accuracy of the Insect-Order and Diptera-Family classification experiments of the Large dataset. The classes are listed in a descending manner with respect to the number of test samples.

## 6 Conclusion

We have described a set of six novel BIOSCAN datasets, on which we conducted image-based classification experiments using the taxonomic annotations of the insects. Looking ahead, iBOL's ongoing efforts will lead to further advancements in several aspects. The rate of insect sample collection is already increasing, resulting in a dataset that is not only larger in terms of the number of records but also more comprehensive, with additional taxa at lower taxonomic levels such as genera and species. Moreover, the dataset will expand to encompass diverse life forms beyond insects. Thus, while the current dataset is already the largest publicly available insect image dataset, it represents just the beginning of what lies ahead.

## Acknowledgement

We acknowledge the support of the Government of Canada's New Frontiers in Research Fund (NFRF), [NFRFT-2020-00073] and a NVIDIA Academic Grant. This research was enabled in part by support provided by Calcul Québec (`calculquebec.ca`) and the Digital Research Alliance of Canada (`alliancecan.ca`). Data collection was enabled by funds from the Walder Foundation, a New Frontiers in Research Fund (NFRF) Transformation grant, a Canada Foundation for Innovation's (CFI) Major Science Initiatives (MSI) Fund and CFREF funds to the Food from Thought program at the University of Guelph. The authors also wish to acknowledge the team at the Centre for Biodiversity Genomics responsible for preparing, imaging, and sequencing specimens used for this study, as well as Utku Cicek for their help with the project.

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

# Appendix

## A1   Data collection and organization

The BIOSCAN-1M Insect dataset consists of insect RGB images and a metadata file containing taxonomic annotation, DNA barcode sequences, and an assigned Barcode Index Number (BIN). In the following sections, we describe the resources available within the dataset.

### A1.1   RGB images

To facilitate different levels of visual processing we created 6 packages of color images of varying sizes. These packages are as follows:

**Original full size RGB images.** The original images are converted to JPEG image format. These images each have a resolution of 2880×2160, and they are typically around 5 MB in size, however some images are smaller at 600–800 kB. The package is structured as 113 zip files, each of which contains 10,000 images except the last (zip file 113 contains 8,131 original full size images). The total size of this package is 2.5 TB. All 113 zip files are stored within the BIOSCAN project space in GoogleDrive as described in Section A2 inside a folder named **BIOSCAN_original_images** and the zip files named as **bioscan_images_original_full_part<n>** where **n** is the partition ID and is in the range of 1 to 113.

**Cropped RGB images.** The images in this package are cropped by our cropping tool as described in the main body of the paper and available in the accompanying BIOSCAN-1M code repository. The package is structured into six zip files where each file contains 20 partitions (20×10,000 files), except the last zip file which contains 13 partitions. The total size of this package is 151 GB. All six zip files are stored within the BIOSCAN project space in GoogleDrive as described in Section A2 inside a folder named **BIOSCAN_cropped_images** and the zip files named as **bioscan_images_cropped_part<m-n>** where **m-n** indicate the start and end partition ID, in the range of 1–113.

**Resized original RGB images.** This package is available in two archive formats (zip and HDF5). The package contains downscaled versions of the original images, requiring reduced storage space. The resizing was done such as to reduce the smaller dimension of image to 256 pixels (and the longer side scaled to preserve the aspect ratio of the original image) and then saved in JPEG format. The total size of these packages are approximately 27 GB, and they are named as **original_256.zip** and **original_256.hdf5**.

**Resized cropped RGB images.** This package is also available in two archive formats (zip and HDF5). The package contains resized versions of the cropped images. The resizing was done such as to reduce the smaller dimension of image to 256 pixels (and the longer side scaled to preserve the aspect ratio of the cropped image) and then saved in JPEG format. The total size of these packages are approximately 7 GB, and they are named as **cropped_256.zip** and **cropped_256.hdf5**.

### A1.2   Metadata file

To enhance the metadata of our published dataset, we incorporated structured metadata following Web standards. The metadata file for our dataset is named **BIOSCAN_Insect_Dataset_metadata**. We created two versions of this file: one data frame in TSV format (**.tsv**) and the other in JSON-LD format (**.jsonld**). The JSON-LD file was validated using the *Google Inspection Tool*.

The metadata file is a table with 22 columns, which contain content as described below. Note that if a sample was not labelled by taxonomist, for each taxonomy ranking group (columns 4–13) the corresponding annotation is listed as **not_classified** instead. Similarly, if a sample has no association with an experiment shown by columns 16–21, then the sample's role is shown as **no_split**.

1. **sampleid**: An identifier given by the collector.
2. **processid**: A unique number assigned by BOLD to each record.
3. **uri**: Barcode Index Number (BIN).
4. **name**: Taxonomy ranking classification label.
5. **phylum**: Taxonomy ranking classification label.

6. **class**: Taxonomy ranking classification label.

7. **order**: Taxonomy ranking classification label.

8. **family**: Taxonomy ranking classification label.

9. **subfamily**: Taxonomy ranking classification label.

10. **tribe**: Taxonomy ranking classification label.

11. **genus**: Taxonomy ranking classification label.

12. **species**: Taxonomy ranking classification label.

13. **subspecies**: Taxonomy ranking classification label.

14. **nucraw**: Nucleotide barcode sequence.

15. **image_file**: Image file name stored in structured packages.

16. **large_diptera_family**: Image association with the training, validation, and test split of experiment-1.

17. **medium_diptera_family**: Image association with the training, validation, and test split of experiment-2.

18. **small_diptera_family**: Image association with the training, validation, and test split of experiment-3.

19. **large_insect_order**: Image association with the training, validation, and test split of experiment-4.

20. **medium_insect_order**: Image association with the training, validation, and test split of experiment-5.

21. **small_insect_order**: Image association with the training, validation, and test split of experiment-6.

22. **chunk_number**: A unique ID to locate the corresponding images within the dataset packages.

## A2 Informational content

The link to access the dataset and its metadata is `https://biodiversitygenomics.net/1M_insects/`.

## A3 Ethics and responsible use

The BIOSCAN project started by the International Barcode of Life (iBOL) Consortium, has collected a large dataset of hand-labelled images of insects. Each record is taxonomically classified by human experts, and accompanied by genetic information.

The publication of BIOSCAN-1M Insect dataset is a common effort made by researchers from the University of Waterloo, Simon Fraser University, Aalborg University, Dalhousie University and the University of Guelph with support from the Vector Institute for Artificial Intelligence, Alberta Machine Intelligence Institute, Pioneer Centre for AI, and the Centre for Biodiversity Genomics.

The availability of the BIOSCAN-1M Insect dataset presents an immense opportunity for scientific advancement and understand in of insect biodiversity. However, it is important to emphasize the ethical and responsible use of this data.

First and foremost, researchers and institutions must prioritize the protection of individuals' privacy and adhere to data protection regulations and guidelines. To our knowledge, there is no personal or identifiable information in the dataset. However, any such information associated with the dataset should be treated with utmost care and reported to the authors.

Furthermore, the researchers and organizations who utilize the BIOSCAN-1M Insect dataset should ensure transparency in their methodologies and practices. This includes clearly stating the purpose of their research, obtaining informed consent when applicable, and maintaining integrity in the interpretation and reporting of the results.

The responsible use of the BIOSCAN-1M Insect dataset entails promoting open collaboration and sharing of knowledge within the scientific community. Researchers should foster an environment

that encourages exchange of ideas, methodologies, and findings, while giving credit to the original dataset creators. It is essential to acknowledge and respect the contributions of the human experts who hand-labelled the images by taxonomically classifying specimens. Proper attribution and recognition should be given to these individuals, as their expertise and efforts are instrumental in the creation and accuracy of the dataset.

## A4    Dataset availability and maintenance

The BIOSCAN-1M Insect dataset and all its content described in the previous sections are available on a GoogleDrive folder named **1M_Image_project**. To access the BIOSCAN-1M Insect dataset, please visit `https://biodiversitygenomics.net/1M_insects/`. We've published a code repository for dataset manipulation, including tasks like downloading dataset packages, image and metadata reading, image cropping, dataset subsampling, partitioning into train, validation, and test sets, and running the classification experiments presented in the BIOSCAN-1M Insect paper.To access the BIOSCAN-1M code repository, please visit `https://github.com/zahrag/BIOSCAN-1M`.

## A5    Licensing

Table A1 shows the copyright associations related to the BIOSCAN-1M Insect dataset with the corresponding names and contact information.

Table A1: Copyright associations related to the BIOSCAN-1M Insect dataset

| Copyright Associations | Name & Contact |
| --- | --- |
| Image Photographer | CBG Robotic Imager |
| Copyright Holder | CBG Photography Group |
| Copyright Institution | Centre for Biodiversity Genomics (email:CBGImaging@gmail.com) |
| Copyright License | Creative Commons-Attribution Non-Commercial Share-Alike (CC BY-NC-SA 4.0) |
| Copyright Contact | collectionsBIO@gmail.com |
| Copyright Year | 2021 |

## A6    Experiment details and results

### A6.1    Backbone models

We utilized two distinct pretrained backbone models for our experiments with the BIOSCAN-1M-Insect dataset. A comprehensive comparison between these models is presented in this section and in Table A2.

ResNet-50 [21] is a deep convolutional neural network, which includes residual blocks that allow for the training of very deep networks without falling into the vanishing gradient problem. ViT-Base-Patch16-224 [61, 17] signifies that the ViT model is designed to process images with a resolution of 224x224 pixels. Each image is divided into smaller patches of size 16x16 pixels, which are then fed into the transformer layers. Each transformer layer includes multi-head self-attention mechanisms and feed-forward neural networks.

Table A2: A comparison between the two pretrained backbone models used in our experiments: ResNet-50 and the ViT-Base-Path16-224. CNN and FC denote Convolutional Neural Network, and Fully Connected layers, respectively.

| Features | ResNet-50 | ViT-B/16 |
| --- | --- | --- |
| Layers | 50 | 12 |
| Based Networks | CNN, Pooling and FC | Transformer |
| Number of parameters | 25.6 M | 86 M |

Overall ResNet-50 has a deeper architecture with more layers than ViT-B/16. This depth can enable it to learn hierarchical features in the data, while ViT's strength lies in capturing relationships between patches by applying self-attention mechanisms, which enables it to capture long-range dependencies in images thus making it suitable for both local and global context understanding. Moreover, due to

its transformer architecture, ViT can parallelize training more effectively, which can result in faster convergence times despite its higher number of parameters.

## A6.2 Validation results

Table A3 shows the performance of all 24 experiments conducted with 3 different seeds using the validation set. According to the validation results, ViT-B/16 with the Cross-Entropy loss function consistently outperforms other models.

Comparing Focal loss to Cross-Entropy, we found that Cross-Entropy produced slightly better results. This could be due to insufficient fine-tuning of Focal loss hyperparameters (alpha and gamma). Furthermore, addressing class imbalance could involve selectively oversampling the less frequent classes during training. This strategy boosts their representation in the training process. For Focal loss, limited exposure to rarer classes might hinder the effectiveness of the re-weighting mechanism.

The presented results of table A3 depict the mean accuracy across various seeds, accompanied by the standard deviation from the average values of each model. The outcomes reveal a notable consistency in the performance of almost all models.

The six models highlighted in bold in Table A3 are used for inference in the test experiments and to report the final results. Pretrained classification checkpoints of these six models, which achieved the best validation accuracy, are available in the GoogleDrive project folder under the directory named **BIOSCAN_1M_Insect_checkpoints**.

Table A3: The table displays Micro-Average-Top-1 and Macro-Average-Top-1 validation accuracy across 24 experiments conducted using 3 distinct seeds. These experiments encompass varying data sizes (Small, Medium, Large), loss functions (Cross-Entropy, Focal), and pretrained backbone models (ResNet-50, ViT-B/16). The experiments utilize consistent hyperparameters and extend to both Insect-Order and Diptera-Family classification levels.

| | | | Micro-Top-1 | | Macro-Top-1 | |
|---|---|---|---|---|---|---|
| Dataset | Backbone | Loss Fn | Insect-Order | Diptera-Family | Insect-Order | Diptera-Family |
| Large | ResNet-50 | CE | **99.65**±0.10 | 97.30±0.02 | 86.26±0.30 | 89.98±0.27 |
| | | Focal | 99.62±0.06 | 97.15±0.00 | 84.66±0.21 | 89.42±0.58 |
| | ViT-B/16 | CE | 99.58±0.21 | **97.67**±0.01 | **87.36**±1.20 | 91.47±0.31 |
| | | Focal | 99.52±0.27 | 97.58±0.02 | 85.80±1.75 | **91.54**±0.21 |
| Medium | ResNet-50 | CE | 98.98±0.04 | 96.24±0.05 | 87.30±1.29 | 91.24±0.33 |
| | | Focal | 98.85±0.04 | 95.92±0.04 | 86.61±0.51 | 90.64±0.22 |
| | ViT-B/16 | CE | **99.14**±0.04 | **96.74**±0.06 | **88.40**±1.17 | **92.83**±0.16 |
| | | Focal | 99.11±0.04 | 96.55±0.02 | 86.75±1.46 | 92.23±0.35 |
| Small | ResNet-50 | CE | 97.79±0.08 | 93.23±0.24 | 87.37±0.56 | 91.43±0.36 |
| | | Focal | 97.62±0.09 | 92.57±0.07 | 86.55±0.60 | 90.68±0.20 |
| | ViT-B/16 | CE | **98.34**±0.10 | **94.46**±0.15 | **88.74**±1.16 | **92.93**±0.33 |
| | | Focal | 98.26±0.03 | 94.42±0.04 | 88.61±0.09 | 92.92±0.16 |

## A6.3 Confusion Matrix

For an in-depth analysis of the performance of models trained under various configurations, we provide detailed Confusion Matrices for the classification experiments conducted at the order and family levels. These experiments were carried out using the model employing ViT-B/16 and the Cross-Entropy loss function. The evaluation was performed on the test set of the Large dataset. You can refer to Figures A1 and A2 for a visual representation of the respective Confusion Matrices.

## A6.4 Qualitative analysis

In this section, we provide a qualitative analysis of the performance results from the order classification experiment on the Small dataset. We aim to shed light on the misclassifications made by our model by visually examining some of the misclassified images.

Surprisingly, roughly 57% of the misclassifications in order-level classification experiments on the Small dataset, using 10,000 test samples, can be traced back to low-quality insect images. This is evident when examining the examples shown in Figure A3, where image quality hampers accurate

Table A4: The table presents the Micro-F1-Score and Macro-F1-Score of our trained models, evaluated on the validation set, and then averaged across different seeds.

| Dataset | Backbone | Loss Fn | Micro-F1-Score | | Macro-F1-Score | |
|---|---|---|---|---|---|---|
| | | | Insect-Order | Diptera-Family | Insect-Order | Diptera-Family |
| Large | ResNet-50 | CE | 99.67±0.07 | 97.44±0.03 | 87.50±0.04 | 90.73±0.23 |
| | | Focal | 99.63±0.06 | 97.28±0.01 | 84.66±0.87 | 90.22±0.26 |
| | ViT-B/16 | CE | **99.68**±0.06 | **97.68**±0.01 | **87.94**±1.59 | **92.01**±0.12 |
| | | Focal | 99.62±0.14 | 97.58±0.01 | 86.98±2.06 | 91.91±0.22 |
| Medium | ResNet-50 | CE | 99.00±0.04 | 96.26±0.06 | 87.43±1.02 | 92.61±0.03 |
| | | Focal | 98.88±0.05 | 95.98±0.05 | 86.77±1.21 | 91.77±0.5 |
| | ViT-B/16 | CE | **99.14**±0.04 | **96.75**±0.04 | **89.33**±1.21 | **93.58**±0.08 |
| | | Focal | 99.12±0.03 | 96.56±0.01 | 87.52±1.01 | 93.20±0.21 |
| Small | ResNet-50 | CE | 97.84±0.11 | 93.27±0.35 | 87.89±0.77 | 92.10±0.51 |
| | | Focal | 97.63±0.04 | 92.78±0.06 | 87.52±0.62 | 91.37±0.12 |
| | ViT-B/16 | CE | **98.31**±0.10 | **94.54**±0.16 | **88.92**±0.74 | **93.56**±0.23 |
| | | Focal | 98.28±0.03 | 94.42±0.06 | 88.36±0.28 | 93.48±0.05 |

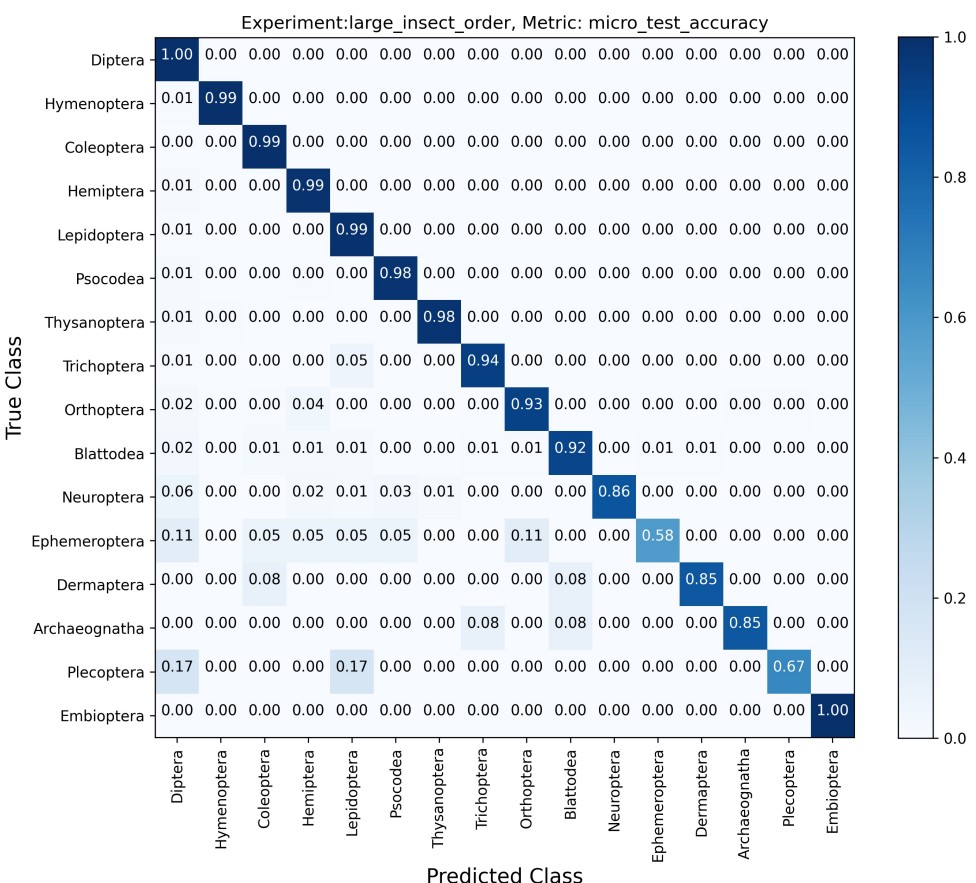

Figure A1: The Confusion Matrix displays the per-class predictions of the order level classification using the Large dataset of the BIOSCAN-1M Insect dataset. The test evaluation is performed on the best model achieved from validation performance results presented in Table A3.

classification. A similar analysis revealed that approximately 45% of the misclassifications in order-level experiments with the Large dataset, using 225,660 test samples, were also attributed to low-quality insect images.

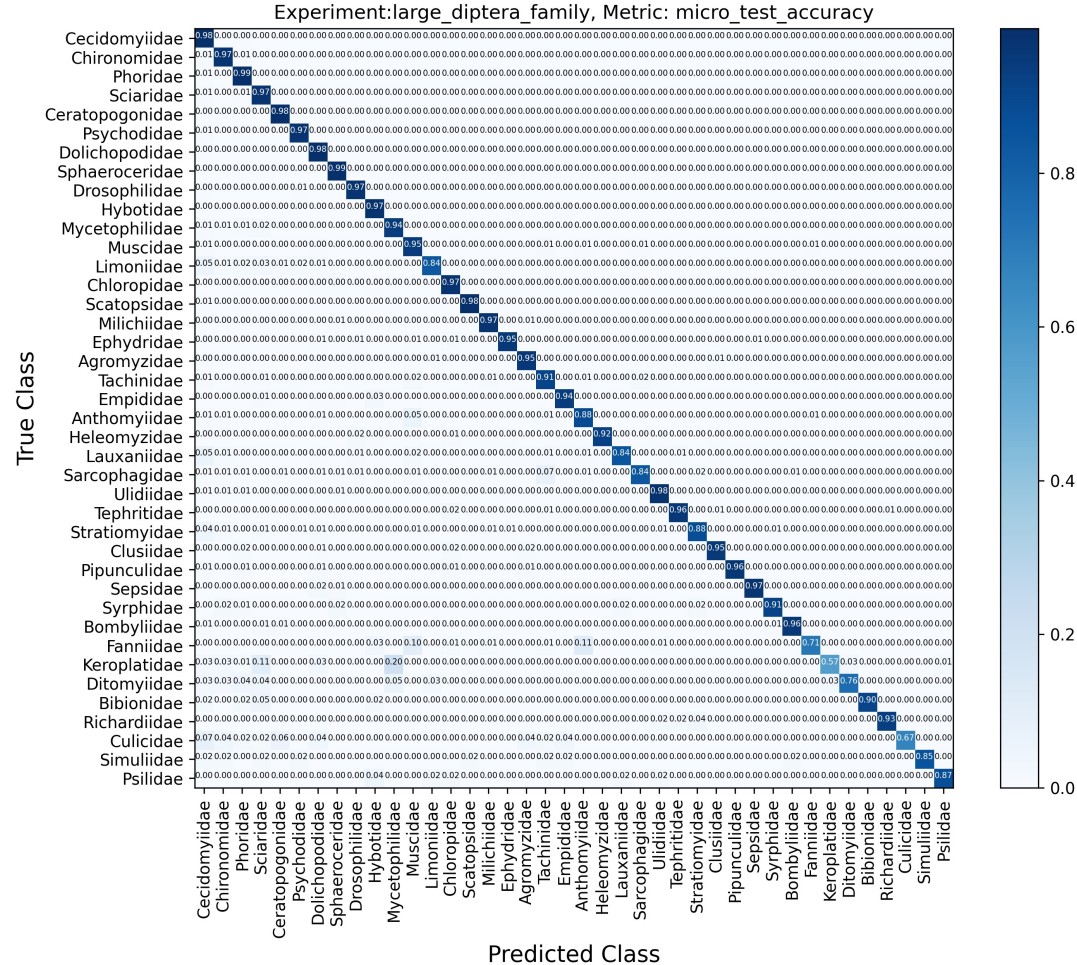

Figure A2: The Confusion Matrix displays the per-class predictions of the family level classification using the Large dataset of the BIOSCAN-1M Insect dataset. The test evaluation is performed on the best model achieved from validation performance results presented in Table A3.

Another observation shows that large proportion of misclassifications are the insects belonging to different orders that are all incorrectly classified as one of the dominant classes of our Small dataset. As an example there are 16.2% of the misclassifications in order-level classification experiments on the Small dataset were insects belonging to different orders are all incorrectly classified as Diptera (flies or mosquitoes), which is the dominant class. This observation, illustrated in Figure A4, highlights specific instances where the model struggles to differentiate between various orders and tends to favour Diptera as the predicted classification.

By examining these qualitative analyses, we gain insights into the challenges faced by our model in correctly classifying insect orders, especially when dealing with low-quality images and distinguishing between similar orders when these orders have low number of training samples.

Our classification experiments have an important application in data cleaning. By identifying low-quality images that have been misclassified, we can effectively detect and remove them from the dataset. This process plays a crucial role in enhancing the overall quality and reliability of the data, as it ensures that only high-quality images of insects are retained.

Furthermore, our classification experiments also enable us to validate the taxonomic classifications performed by human experts. By examining instances of false predictions, we can investigate

whether a sample has been incorrectly annotated, providing valuable insights into the accuracy of the taxonomic classification process.

### A6.5 Discussion

#### A6.5.1 Dataset: Generation, Curation and Growth

The BIOSCAN project is currently in its initial stages, with its primary goal being the facilitation of a global biodiversity assessment. In this section we clarify certain aspects and procedures accomplished in the generation of the BIOSCAN-1M Insect dataset.

All samples of the BIOSCAN-1M Insect dataset were processed at one facility using the same workflows and imaging equipment. This should exclude all potential biases with respect to data collection.

Regarding dataset labeling procedure, the order-level classification is conducted by taxonomists and entomologists primarily relying on morphology rather than barcode matching. For family and finer-grained classifications, a combination of approaches may be employed, often supported by BIN assignment.

The annotators responsible for labeling the data were personnel affiliated with the Centre for Biodiversity Genomics, including technicians and research scientists, all engaged in full-time roles at the institute and receiving equitable compensation. A diverse team of approximately 15 to 20 individuals participated in providing labels for the dataset. These labels were derived from the images, forming the foundation for establishing higher-level taxonomies such as order and family. Additionally, the annotators conducted visual examinations of specimens for finer-grained taxonomic ranks, utilizing primary literature as a reference where feasible. Taxonomic experts (human professionals) engaged in the barcoding process have varying level of involvement, which tends to increase when the placement of a specimen is more contentious. Additionally, these experts are responsible for describing species, a task not handled by the machine. They also supply the reference ID that enables us to establish matches.

Several factors contribute to why most samples of the dataset are classified only to the family and finer-grained classes are not provided, with the primary one being the time constraints associated with completing the assignment. The complexity of the task arises from the fact that relying solely on a barcode is often insufficient due to potential ambiguities, as discussed earlier. Each sample's labeling requires verification through visual inspection (images), or in some cases, examination of the original specimen, before proceeding with further classification. This process is not easily scalable, prompting the adoption of BINs as a species proxy.

However, the process utilized to expand the dataset remains consistent with the methodology employed for the BIOSCAN-1M Insect dataset. The Dataset will be retrained at regular intervals and older versions archive stored on Zenodo and date stamped. Similarly we will use GitHub's releases mechanism to version the accompanying code.

#### A6.5.2 Application: Model and Tasks

In this article the baseline problems and methods we explored were chosen to be simple and accessible and as a result, limited. They are not the focus of the paper as our primary focus is to release the dataset and showcase its inherent potential. We expect future works will use the dataset for interesting problems such as hierarchical classification, zero-shot classification, set-valued classification and methods that improve performance in the fine-grained and long-tailed label regime.

We believe that the most promising methods will be hierarchical classifiers that yield uncertainty estimates over multiple taxonomic levels. Improving performance on minority classes and reliably delineating novel operational taxonomic units is also important. To get there, we believe the most promising areas of investigation from the ML side will be semi-parametric methods that use reference libraries at test time, set-valued classification as a natural means of expressing uncertainty, and zero-shot classification.

The utilization of the BIOSCAN-1M Insect dataset in conjunction with other large biological datasets from variuos domains becomes feasible by harnessing the preprocessing module proposed in this paper. By employing tools like our cropping tool and applying machine learning techniques for domain adaptation/generalization, one can capitalize on the capabilities of a pre-trained model on

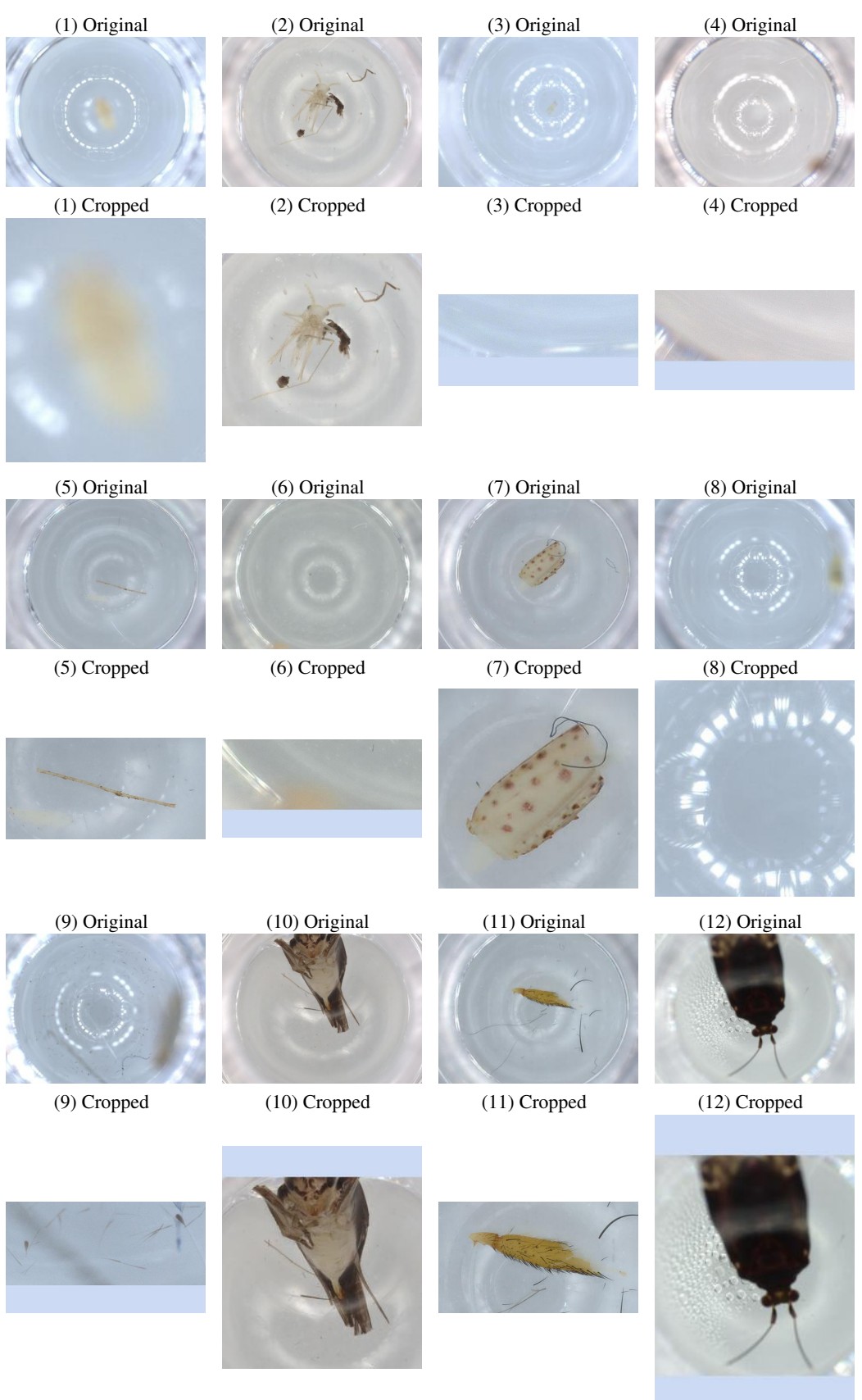

Figure A3: Examples of misclassifications caused by low quality images photographed from insects.

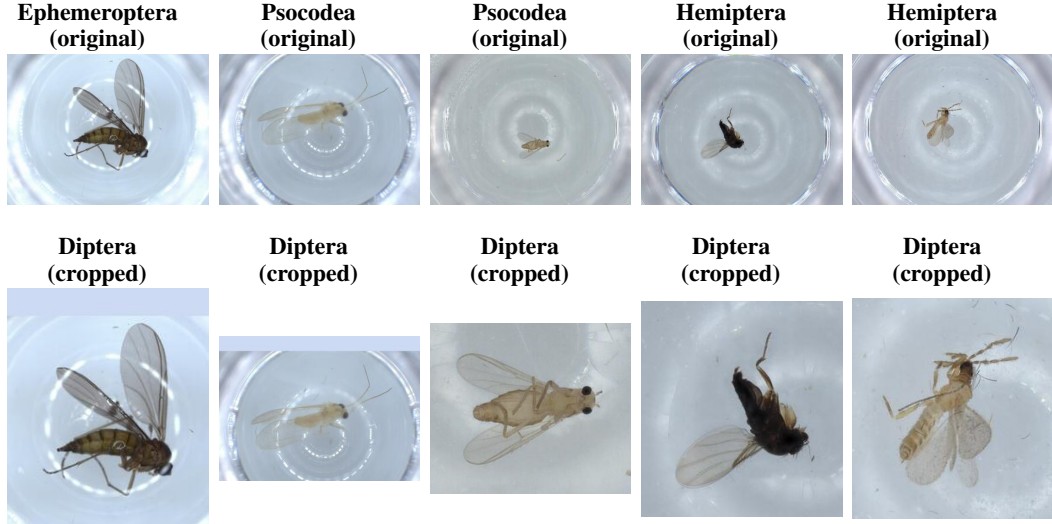

Figure A4: Examples misclassified as the dominant class Diptera (flies).

BIOSCAN-1M Insect images to effectively tackle classification challenges in out-of-distribution scenarios.

Overall, we believe that the unique annotation and metadata including the DNA barcodes will prompt interesting multimodal strategies. We intend to enhance our approach by incorporating DNA barcode sequences and utilizing Barcode Index Numbers (BINs). This strategic direction aims to effectively tackle the limitations associated with the current taxonomic labels of the images. Notably, the utilization of BINs holds promise as each image is inherently associated with a distinct and unique BIN.

## A7    Preprocessing: Cropping tool

Our observations showed significant improvement in processing time when we used cropped images rather than original ones. However, cropping is a challenging problem since insect images have varying shapes, sizes, colors which is also shown in Figure A5. The illumination and background color and surface are not the same across the original images.

Furthermore, there are cases in the original images that the insect is photographed in pieces and in such cases the cropping is quite challenging especially when the insect is small, and its less discriminative body parts like legs are distant from the main body so these pieces could be cropped instead.

To address these issues more effectively, we have developed a tool based on the DETR model for automatic identification and cropping of the main insects in images. The primary objective of this tool is to facilitate data storage and subsequent research, such as neural network training. The tool uses the DETR model to accurately locate the main insects in images and crop accordingly. By removing irrelevant background information, the tool optimizes storage space and reduces the time spent on data management. Additionally, the cropped images can be effectively used for tasks such as image classification through neural network training, leading to improved performance in the following image classification task. Our crop tool checkpoint is available in the GoogleDrive project folder under the directory named **BIOSCAN_1M_Insect_checkpoints**.

### A7.1    Approach

The cropping tool consists of first detecting a tight bounding box for the insect in the image using an object detector and then cropping the image by extending the bounding box. We show an overview of the cropping tool in Figure A6. To accurately locate the insect in the image, we chose the DETR [9] model which has excellent performance in the task of object detection and the corresponding pre-trained ResNet-50 [21] as the feature extractor. At the beginning, the CNN-based feature extractor extracts a set of image features that are fed into a transformer-based encoder-detector. The detector takes a set of learned positional embeddings as object queries and uses them to attend to the encoder

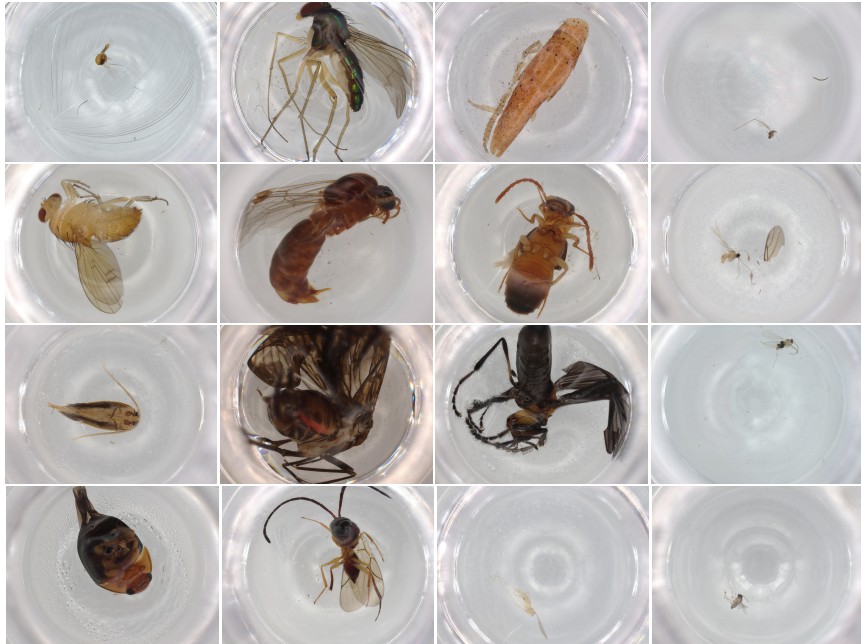

Figure A5: Examples of images used to adapt our cropping tool. We include variations of insects' size, color, position and shape.

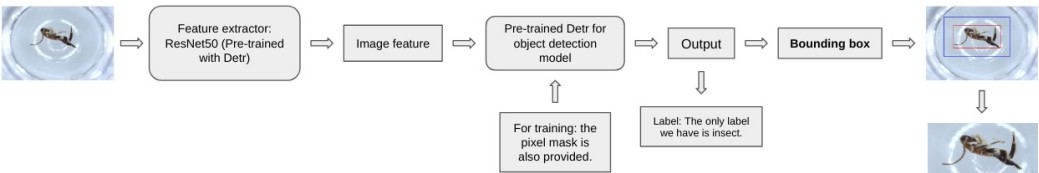

Figure A6: Our DETR [9] based cropping tool takes an input image, extracts features using a ResNet50 [21] backbone, and extracts a tight fitting bounding box for the insect (see red box). We then extend the bounding box (see blue box) to obtain the final cropped image. We use a DETR model pretrained on MSCOCO [35]. To fine-tune the DETR model, we annotate a small set of insect images with their segmentation mask.

outputs. Each of the output decoder embeddings is then passed to a shared FFN which will predict whether there is "no object" or a detected object with its class and bounding box. Each bounding box is parameterized as $(cx, cy, w, h)$ where $(cx, cy)$ is the center of the bounding box, and $(w, h)$ is the width and height of the box, all normalized to 1.

The DETR network is trained by optimizing a bipartite set loss that matches detected boxes with the ground-truth boxes using the Hungarian algorithm to minimize the overall matching loss between the matched pairs. The pairwise matching loss is a combination of the classification loss and a box regression loss (the bounding box loss is included only when the detected box matches a ground truth box that corresponds to an object, and is a weighted combination of GIOU [48] and L1 loss between the bounding box parameters). In our case, we have only one object class ("insect") so the classification reduces to a binary classification between "insect" and "no object".

Note that other than the ground-truth bounding box, for training the DETR model of the cropping tool, the pixel mask of the insect in the image is also required for the training. This pixel mask is not needed during the inference phase.

**Training details.** We start with a DETR model pretrained on MSCOCO [35] and fine-tune it on our dataset. We use the AdamW [38] optimizer with learning rate of 0.0001, weight decay of 0.0001 and a batch size of 8. We train for 10 epochs. On a RTX 2080 Ti with 4 workers, for 1,000 images, training takes 1.5 minutes per epoch and a total of 15 minutes for 10 epochs.

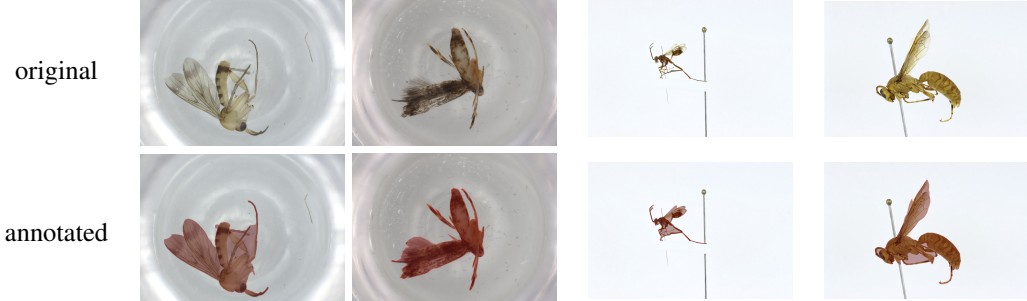

Figure A7: Typical instances of annotated IW (left two columns) and IP (right two columns) images. To obtain an accurate bounding box in reasonable annotation time, we focused on drawing the external outline of the main insect only excluding the small spaces between its legs. Small parts of the insect that are far away from the main body (e.g. the small leg in the first image) are also not included.

The original DETR is trained with images resized to fit within an 800 ×1,333 tensor. We match that and resize our image (preserving the aspect ratio) so that the shortest side is less than 800 and the longest side is less than 1333. No data argumentation is applied during training.

**Cropping.** In the cropping phase, With the predicted bounding box (the red bounding box in Figure A6), we can choose to enlarge it using a certain method to include more details or meet specific image aspect ratio requirements. By default, we will choose 0.4 times the longest edge as the target and extend this size in both height and width to produce the final cropping bounding box (the blue bounding box in Figure A6).

To crop the image, we run our fine-tuned DETR model on the input image to identify the tight bounding box around the insect. We assume that each image contains one insect of interest, and during cropping, we take the predicted bounding box with the highest probability that is higher than 0.5. Before cropping, we extend the predicted bounding box by a fixed ratio $R = 1.4$ of the size of the tight bounding box. We extend the height and width by the same number of pixels by computing the extended size as: $\text{ExtendSize} = (R - 1) \times \max(\text{width}, \text{height})$.

If the bounding box is at the edge of the original image, we pad the image by adding pixels of maximum intensity to match the white background. In this way, even if the predicted bounding box does not encompass all the details of the insect, we can still include the entire insect in the cropped image. Furthermore, this maintains a more square aspect ratio, which facilitates downstream tasks such as image classification.

**Runtime.** The cropping tool can be run in CPU or GPU mode. On a Linux machine with 16 cores and running 4 workers, using CPU only, 10 k images can be cropped in 2 hours and 40 minutes (images loaded and written to local SSD). Using an RTX 2080 Ti GPU, 10K images can be cropped in 30 minutes on the same machine.

### A7.2 Data

We develop our tool on two sets of images of insects that are pinned (INSECTS-PINNED) and insects in wells (INSECTS-WELL). Using the Toronto Annotation Suite (TORAS) [29], we annotate each with their segmentation mask. For each set, we annotated a large (1,000 images) and a small (100-150 images) training set and another small set for evaluation. The annotation was done by three volunteers and took a total of 4 hours for 1,000 images. The two sets of images are described below (see Figures A7 and A8 for example images and annotated masks):

**INSECTS-PINNED (IP)**. The insect is pinned in these images (or has a pin near it) with a fairly clean white background. The images are taken by a Digital SLR camera (Canon) mounted on a motor-drive positioning system (OpenBuilds ACRO) equipped with stepper motors and a motion control system. Pinned specimens are arrayed in sets of 96 (8 ×12 array) in a large enough distance between them to avoid including parts of neighbouring specimens in the image frame. For this set, we collected 1,000 images to form the large training set (IP-1000-train), 100 images for the small training set (IP-100-train), and another 100 images for the validation set (IP-100-val).

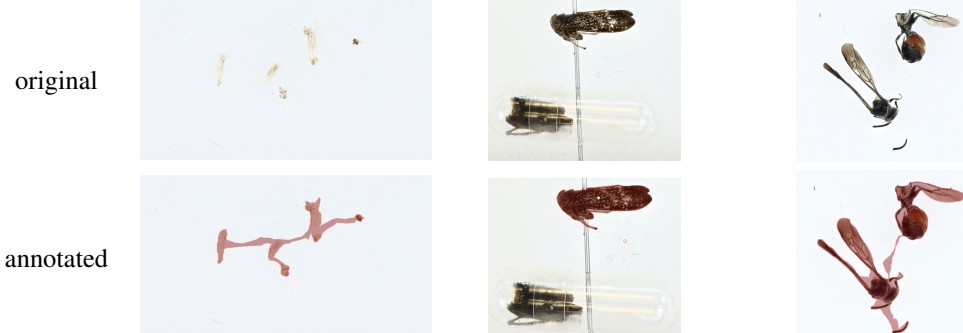

Figure A8: Examples of special annotation cases. Left: for an insect that is broken into multiple parts with even size, we create a mask that covers all of the parts of the insect. The ideal mask should contain minimal background, and keep the edge of the mask as close to the insect's edge as possible (left, right). Middle: for two insects where one is in the container and the other is not, we annotate the insect that is not in the container. Right: for a split insect we annotate all parts.

Table A5: The Average Precision (AP) and Average Recall (AR) were computed on the IW-150 val and IP-100-val datasets using the DETR model, which was pre-trained with different training splits.

| | INSECTS-PINNED-100-Val | | INSECTS-WELL-150-Val | |
| Training data | AP[0.75] | AR[0.50:0.95] | AP[0.75] | AR[0.50:0.95] |
| --- | --- | --- | --- | --- |
| IP-100 | 0.910 | 0.893 | 0.543 | 0.729 |
| IP-1000 | 0.949 | **0.918** | 0.415 | 0.587 |
| IW-150 | 0.415 | 0.587 | 0.801 | 0.802 |
| IW-1000 | 0.665 | 0.695 | 0.872 | 0.835 |
| IP-1000 + IW-1000 | **0.964** | 0.907 | **0.901** | **0.885** |

**INSECTS-WELL (IW).** In these images, the insects are placed in a well. Here the images tend to have a less clean background due to the glass and uneven reflected light. The images are taken using a Keyence VHX-7000 Digital Microscope system with a fully integrated head and automatic stage that permits high-resolution (4 k) microphotography of individual specimens. Because its scanning stage can hold a 96-well plate, the system automatically acquires a high-resolution image of each specimen by controlling movements in the X-Y plane. As well, its capability to control the z-axis position of the stage with a precision of 0.1 m allows it to photograph each specimen at multiple heights before rapidly compiling these images into an in-focus image (depth stacking). For this set, we collected 1,000 images to form the large training set (IW-1000-train), 150 images for the small training set (IW-150-train), and another 150 images for the validation set (IW-150-val).

Note that the BIOSCAN-1M Insect Dataset consists only of insects in wells. We include the insects with pins to extend the usefulness of the cropping tool for a broader spectrum backgrounds that may appear in the process that specimens are acquired in the larger BIOSCAN project.

During annotation, we focus on masking the main insect and we exclude small broken pieces of the insect that are far from its body (see Figure A7). There are also challenging cases where the insect may be broken into pieces or there are multiple insects (see Figure A8). For insects that are broken into multiple pieces of similar size, we create a mask that covers all the pieces. When there are multiple insects, we mask only the central insect.

### A7.3  Experiments

### A7.3.1  Metrics

The metrics we used are the Average Precision (AP) and the Average Recall (AR) with the IOU of the bounding box equal to [0.75] and [0.50:0.95], as they measure the precision and recall aspects of detection performance. AP reflects the accuracy of detection by considering the overlap between predicted and ground truth bounding boxes, while AR assesses how well the system captures all the ground truth objects.

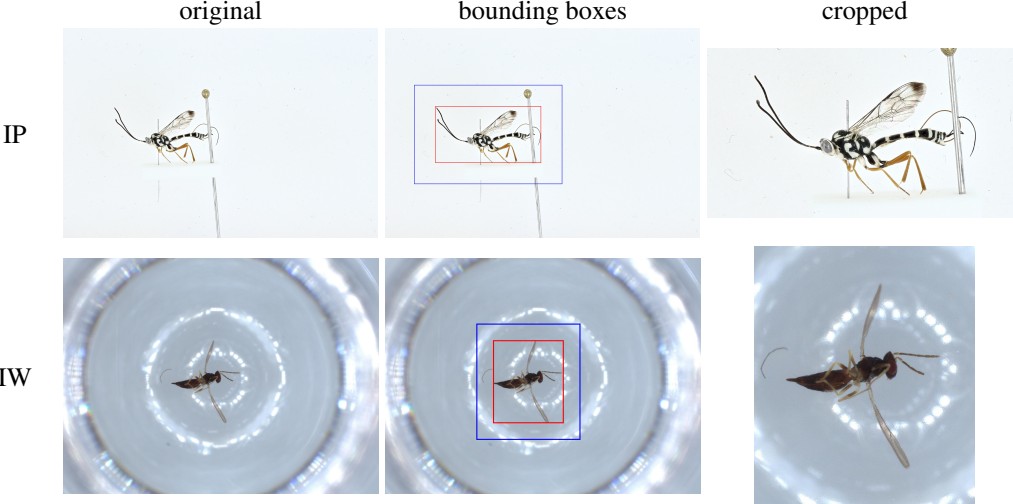

Figure A9: Cropping examples of images from INSECTS-PINNED (IP) and INSECTS-WELL (IW) with the original image, image with detected bounding boxes in red, extended bounding boxes in blue, and final cropped image.

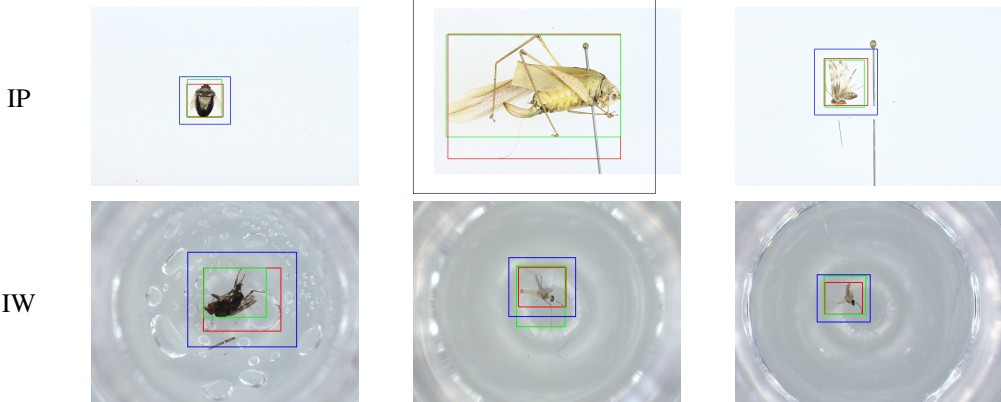

Figure A10: Examples of imperfect insect detection (IOU < 0.85), with ground-truth bounding box in green, detected bounding boxes in red, and extended in blue. In the second image of IP, note that we extend the image with the white background to fit the bounding box that escapes the original image boundaries.

### A7.3.2 Cropping results

We show examples of cropped images in Figure A9. The images show the accurate identification of the insect subject by the DETR model (red bounding box) and the extended bounding box (blue bounding box) used for cropping. In Figure A10, we show cases where the predicted bounding boxes have an intersection over union (IoU) with the ground truth bounding boxes (green bounding box) less than 0.85. From these examples, we observe that the antennae of certain insects and the presence of cluttered backgrounds sometimes can create disturbances to our fine-tuned DETR model. However, by expanding the predicted bounding boxes, we are still able to capture all the desired information within the cropped images.

To evaluate the performance of our cropping tool with different amount and type of data, we trained the DETR model with 5 training splits (IP-100, IP-1000, IW-150, IW-1000 and IP-1000+IW-1000), and evaluate these models on two validation splits(IP-100-val and IW-150-val). Overall, from Table A5, we see that using the mixed training split with 1000 images from IP and 1000 images from IW results in the highest accuracy. This is the model that we use for cropping the images in the BIOSCAN-1M Insect Dataset.

Table A6: Comparison of classification accuracy results on original images vs. cropped images. Both are resized to 256 on the smaller dimension. Overall, we find the cropped images yield slightly higher accuracy.

| | Order-level | | | | Family-level | | | |
| | Micro-average | | Macro-average | | Micro-average | | Macro-average | |
| Image type | Top-1 | Top-5 | Top-1 | Top-5 | Top-1 | Top-5 | Top-1 | Top-5 |
|---|---|---|---|---|---|---|---|---|
| original | 0.9626 | 0.9970 | 0.8218 | 0.9964 | 0.9248 | **0.9802** | 0.9109 | **0.9730** |
| cropped | **0.9786** | **0.9976** | **0.8757** | **0.9980** | **0.9314** | 0.9786 | **0.9154** | 0.9728 |

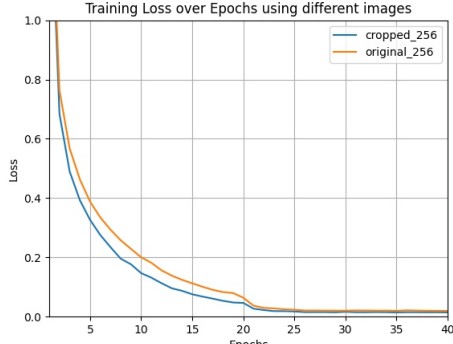 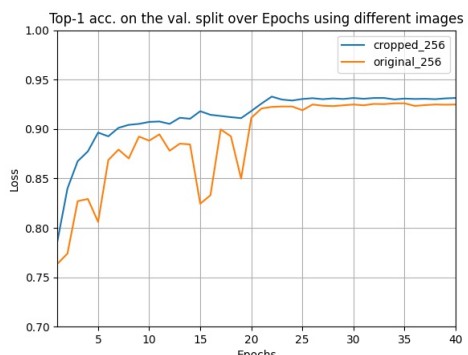

Figure A11: The training loss and Top-1 accuracy on the validation split during the training of family-level classification of images of insects using cropped (blue) and original (orange) images. Both are resized to 256 on the shorter side.

### A7.3.3 Insect classification using cropped images

We further evaluate the effectiveness of our auto-cropping tool on a downstream task: insect image classification at the order/family level. In Table A6 we compare the classification performance of the original vs. cropped images on the BIOSCAN small dataset following the training setup we described in the main paper. We use the ResNet-50 backbone with cross-entropy loss and train with the AdamW optimizer with a learning-rate of $0.001$ and momentum of $0.9$ for 100 epochs for order-level classification and 40 epochs for family-level classification. All images are resized such that the shorter side has size 256. During training, we apply random horizontal flip with probability of 0.5, and random crops of $224 \times 224$ are extracted and fed into the backbone to extract image features. During inference, the center $224 \times 224$ crop is extracted. We measure the micro and class macro average top-$K$ accuracy at $K = 1$ and $K = 5$.

From Table A6, we see that in most cases, using cropped images to perform training results in higher classification accuracy. In the cases where original image type outperforms cropped type, the difference is small.

To further compare the difference between using original images and cropped images for training, we also compare the loss curve during training with original and cropped images.

By comparing the loss at epoch 10, 15 and 20, we see that using the cropped images can help the model converge faster. Using the cropped images also yields higher top-1 accuracy on the validation split.

