# OpenReview forum: "A Step Towards Worldwide Biodiversity Assessment:  The BIOSCAN-1M Insect Dataset"
_NeurIPS.cc/2023/Track/Datasets_and_Benchmarks — NeurIPS 2023 Datasets and Benchmarks Poster_

### Official Review · Reviewer_4KRL · 2023-07-21
**Very useful dataset for insect identification and classification**

**Rating:** 6
**Confidence:** 3
**Clarity:** The paper is very well written and co…

**Strengths:**

This dataset is very substantial and represents many hours of work needed by experts to carefully annotate and collate. That in itself is very useful for anyone working in this domain and a significant addition to the body of work for taxonomy based classification.

**Additional Feedback:**

Regarding Fig:S1, are there plans to filter the poorly photographed images from the datasets?
To reiterate previous points raised above:
Also I would have liked to see some measure of the consistency of the classification,  and it would be good if the authors is able to provide some classification results using different network sizes.. ie,  (ResNet18, and ViT-L), and on the more challenging task of classifying to a larger number of classes (i/e Name)

**Correctness:**

I would have liked to see some measure of the consistency of the classification results over multiple runs (perhaps 10) and provide a measure of the variance in the classification accuracies.

**Documentation:**

Other than the annotation process, there is sufficient detail on the dataset generation.

**Ethics:**

No I do not suspect there are any ethical concerns with this submission

**Limitations:**

I might have missed it, but i couldn't find a discussion on the limitations of the dataset. In particular, I was hoping there was discussion on the annotation process (whether it was single vs multiple experts), validation of the annotated labels. Hopefully the authors can clarify more on this in the response.

**Opportunities For Improvement:**

Providing the Top-10 classification result for a 16 class classification task (especially when the top 5 accuracies are already relatively high)  seems superfluous. I would have preferred if the authors provided benchmarks for more challenging tasks i.e. Genus/Subfamily on the subset of images with those labels, and name on the major/minor subsetted images. It will give us a better picture on the difficulty of the task and potentially reveal some of the further work needed for vision transformers and conventional CNNs for classifying images in this domain.

Personally I would also like to see more architectures and/or sizes of ResNets/VIT used in the benchmarks which might help reveal the architecture limitations of those networks.

It would also be helpful to have the confusion matrix in a supplementary material.

**Relation To Prior Work:**

The authors relate  this work to other taxonomy image datasets, and a comparison of the scale of the dataset vs coverage and provided a summary on the difference between their work and prior work, namely in the scale of the dataset as well as how t

**Summary And Contributions:**

This paper describes BIOSCAN-1M a large scale high-resolution image dataset of insects in laboratory setting, labeled with taxonomy data (primarily order and family) and some benchmark results using conventional CNN/Tranformer based image classifcation

---

> ### Author Response · Authors · 2023-08-20
> **Rebuttal by Authors**
>
> The authors value the reviewer's evaluations and greatly appreciate their insightful comments.
>
> - We showed Top-10 classification results mainly for Family level classification where there are 40 classes, however, we agree with the reviewer that these results are not informative for our classification experiments and have removed them from Table 4 of the revised manuscript. We fully agree with the reviewer's viewpoint regarding the significance of establishing a Benchmark task for the Species level. Nevertheless, the primary reason that prevented us from pursuing this approach was the restricted availability of detailed fine-grained information at the Species level within the current version of the dataset. Based on the statistical analysis of the BIOSCAN_1M_Insect dataset, we observe that at the Genus and Species taxonomic levels, there exist 3,441 and 8,355 distinct subcategories, respectively. However, the available data samples for classification are relatively constrained, with only 254,096 samples classified at the Genus level and 84,397 samples at the Species level, all of which have been meticulously annotated by experts. This disparity between the considerable number of classes and the limited dataset size poses a significant and interesting challenge for effectively training and optimizing classification models.  We believe this would be a good direction to explore.  Please see our response to R-t14s for a short discussion on possible approaches and also please see Table 1 of the revised manuscript and a Discussion presented in the Supplementary material Section S6, which addresses the limitations exist with the current version of BIOSCAN-1M-Insect dataset.
>
> - Thank you for the useful suggestion. In the Supplementary material (Section S6) we added and discussed more details of our experiments designs and settings as well as the two pretrained backbone models. We have also added the confusion matrices to the Supplementary Material Section S6.4.
>
> - We acknowledge the reviewer's observation concerning the absence of a discussion regarding the potential limitations of the dataset. In response, we have included a dedicated table (Table 1),  and elaborated on the limitations of the BIOSCAN_1M_Insect dataset in the revised version of the paper.
>
> - With respect to the annotation process, the experts responsible for labeling the data were personnel affiliated with the Centre for Biodiversity Genomics, including technicians and research scientists, all engaged in full-time roles at the institute and receiving equitable compensation. A diverse team of approximately 15 to 20 individuals participated in providing labels for the dataset. These labels were derived from the images, forming the foundation for establishing higher-level taxonomies such as Order and Family. Additionally, the annotators conducted visual examinations of specimens for lower taxonomic ranks, utilizing primary literature as a reference. Please see our revision added to the Supplementary material Section S6.6.1.
>
> - We greatly appreciate the valuable suggestions offered by the reviewer. In response to this insightful recommendation, and considering our time and resource limitations, we have conducted our experiments using three distinct seeds. The validation outcomes of these experiments have been comprehensively outlined in the revised Supplementary material, specifically in Section S6.3 (Tables S4 and S5) for our validation set and Table 4 of the revised manuscript for our test set.
>
> - Yes, in the updated version of the dataset the low quality images will be removed. We acknowledge the presence of various research ideas that could be explored using the BIOSCAN_1M_Insect dataset, aligning with our future plans. However, in the context of the current article, our primary focus is to release the dataset and showcase its inherent potential. To accomplish this, we include baseline experiments that illustrate its application with advanced machine learning models. Note that "Name" is not considered as a defined taxonomic ranking group and it hosts the name of the lowest known rank entry, so for instance, if there exists a Species name of an organism, "Name" will show that, but if the organism only has Family level ID, then "Name" will show the ID.
>
> - A more purposeful task would be to predict the BIN from the images, as the BIN is a species-level identifier and we have a BIN label for each image. However, as this would be a classification task with 91k classes, this is no small engineering feat(!) and was beyond the scope of what we could prepare to accompany the release of the dataset. Please see Table 1 and Table 2 of the manuscript for more clarifications..

---

> > ### Comment · Reviewer_4KRL · 2023-08-30
> > **Thank you for the response**
> >
> > Thank you for the response to my review. All in all, I agree that this is a strong paper and should be accepted

---

### Official Review · Reviewer_t14s · 2023-07-23
**Review of BIOSCAN-1M Insect Dataset**

**Rating:** 7
**Confidence:** 4
**Correctness:** To my knowledge, the submission is co…
**Clarity:** The paper is very clearly written and…

**Strengths:**

- The dataset is very large and contains substantial metadata for most samples that can enable researchers to pursue a wide variety of new methods that take advantage of metadata or the hierarchical structure. The labels should be very high quality based on the procedures for annotation described in the paper.
- The authors did a good job of explaining the necessary domain background for understanding the dataset while also providing sufficient detail for ML practitioners to understand how to use the dataset.
- The authors made thoughtful decisions about dataset splits and subsetting that will make the dataset (and its variants) much easier to use for ML practitioners and accessible for a wide variety of users and compute constraints.

**Additional Feedback:**

- Are there any factors in the data collection or sampling that could result in correlation between samples that could potentially bias a model or make the test metrics insufficient for measuring real world OOD performance? (for example, subgroups of the dataset collected by different labs that may share some similarities?)

**Documentation:**

The Zenodo link does not work (Page not found) and I did not find any other links to the dataset and documentation. The authors also stated that the dataset would grow in the future as more samples are collected, but did not mention how this would be managed (e.g. version control for past results).

**Ethics:**

I did not identify any ethical concerns.

**Limitations:**

- The authors did not discuss the potential limitations of the dataset nor potential negative societal impacts (though I can't imagine any potential negative societal impacts from it).
- It seems like there is opportunity for future research using the proposed dataset to focus on specific aspects or subtasks of the dataset that are more challenging or relevant for domain scientists, which I think could have been discussed more. For example, should users of the dataset focus on improving minority class accuracy? Should they optimize for 1 shot or few shot classification accuracy? More discussion of this could help increase the impact of the dataset (even better if these were reflected in the benchmark tasks).
- Minor suggestions:
  - It wasn't clear in Figure 4 if the images are from the original image dataset, or cropped, or resized, etc.
  - Line 233 uses the acronym "FFN" which was not previously defined (I assume this is meant to be feed forward network)

**Opportunities For Improvement:**

- Two benchmark tasks were presented for classifying insect Order and Family, but not more fine-grained classes like Species as other datasets have done. I assume this is because not all samples have the fine-grained class information, but it seems like it would still be valuable to have a benchmark task using the data subset that does have species information?


**Relation To Prior Work:**

The paper gave a good concise summary of previous datasets related to the proposed dataset, but could have more specifically addressed how the proposed dataset fills a need that is yet unmet.

**Summary And Contributions:**

The authors present the BIOSCAN-1M Insect Dataset which contains RGB images, taxonomic classifications, and genetic information for 1 million insects collected from Costa Rica, Canada, and South Africa, constituting the largest insect image dataset published yet. The dataset is published with dataset splits and subsets for two challenge tasks: order classification into 16 orders and family classification into 40 families, both of which exhibit long tail class distributions. Baseline results are presented for deep learning models with different backbones and loss functions for top-k classification accuracy, showing very high accuracy for high-frequency classes but lower accuracy for minority classes.

---

> ### Author Response · Authors · 2023-08-20
> **Rebuttal by Authors**
>
> The authors express their gratitude to the reviewer for their invaluable comments and insightful recommendations.
>
> - We fully agree with the reviewer's viewpoint regarding the significance of establishing a Benchmark task for the Species level. Nevertheless, the primary reason that prevented us from pursuing this approach was the restricted availability of detailed fine-grained information at the Species level within the current version of the dataset. Based on the statistical analysis of the BIOSCAN_1M_Insect dataset, we observe that at the Genus and Species taxonomic levels, there exist 3,441 and 8,355 distinct subcategories, respectively. However, the available data samples for classification are relatively constrained, with only 254,096 samples classified at the Genus level and 84,397 samples at the Species level, all of which have been meticulously annotated by experts. This disparity between the considerable number of classes and the limited dataset size poses a significant challenge for effectively training and optimizing classification models. We have made clarification regarding the limitations with the current version of the dataset in the Table 1 of the manuscript and additional discussion is presented in the Supplementary material S6.6.
> (2) We acknowledge the reviewer's observation concerning the absence of a discussion regarding the potential limitations of the dataset. In response, we have included a dedicated table (Table 1) and elaborated on the limitations of the BIOSCAN_1M_Insect dataset in the revised version of the paper, and additionally made discussion in the Supplementary material Section S6.
>
> - We thank the reviewer for their feedback regarding the opportunity for future research using the proposed dataset to focus on specific aspects or subtasks that are more challenging or relevant for domain scientists. We completely agree. For domain scientists, we believe that the most promising methods will be hierarchical classifiers that yield uncertainty estimates over multiple taxonomic levels. Improving performance on minority classes and reliably delineating novel operational taxonomic units is also important. To get there, we believe the most promising areas of investigation from the ML side will be semi-parametric methods that use reference libraries at test time, set-valued classification as a natural means of expressing uncertainty, and zero-shot classification. Overall, we believe that the unique annotation and metadata including the DNA barcodes will prompt interesting multimodal strategies. In this initial work, it was beyond the scope for us to investigate all these different possibilities.  We hope to investigate some of these directions and to enable the community to explore these interesting avenues with our dataset. We have added this summary to the conclusion. Please see the discussion added to the Supplementary material S6.6.2.
>
> - Figure (4) presents the original images, and FFN is Feed Forward Network.
>
> - We apologize for the issue with the Zenodo link and we thank the reviewer for noticing and pointing this out. We have also checked the links provided in the Supplementary material and verified that these were correct, and added the correct link to the revised manuscript.
>
> - The process utilized to expand the dataset remains consistent with the methodology employed for the BIOSCAN-1M-Insect dataset. Datasets will be retrained at regular intervals and older versions archive stored on Zenodo and date stamped. Similarly we will use GitHub’s releases mechanism to version the accompanying code. We will mention this in the revised paper. Please see our revision in the Supplementary material S6.6.1.
>
> - All samples were processed at one facility using the same workflows and imaging equipment. This should exclude all potential biases with respect to data collection. Please see our revision in the Supplementary material S6.6.1

---

> > ### Comment · Reviewer_t14s · 2023-08-21
> >
> > Thanks for clarifying these points!

---

### Official Review · Reviewer_Dp7Q · 2023-07-25
**A large dataset of insect images, with some clarification required prior to acceptance**

**Rating:** 6
**Confidence:** 4

**Strengths:**

The motivation for the paper is compelling and it is well-written with illustrations of concepts that might not be familiar to the machine learning community (for example taxonomic classification). The dataset itself is large and well-documented, though I have noted some improvements and concerns below.

The paper is accompanied by a [public](https://github.com/zahrag/BIOSCAN-1M) code repository that contains analysis and model training code. While I noted that there were several recent commits (post-submission), they are largely cosmetic fixes and that the repository reflects the manuscript. The code I reviewed looked clean, well structured and well documented.

**Additional Feedback:**

On the whole I appreciate the amount of work that has gone into the paper and, no doubt, in collecting and curating the imagery and sequencing data. I have pointed out some concerns that I have with the way the dataset and analysis has been reported. Were these issues remedied, I would likely score the paper higher.

Please see individual review sections for questions/requests. The general theme of the feedback is improved clarification especially as many readers of the paper will not be intimately familiar with biological data, though I think the authors have done a credible job of making the background understandable to non-experts.

I don't believe that any of the analysis suggestions require new experiments. Assuming the model results are available to the researchers, the requests are a matter of calculating and reporting some additional statistics, and some clarifying remarks.



**Clarity:**

Subject to my concerns in other review sections, I think the paper is well written.

**Correctness:**

- In general I think the submission is correct, aside from clarity around dataset generation and label provenance. I see the main (standalone) contribution to be the curation/collation of the dataset, which presumably took considerable effort and I believe is valuable for the community. The model section is weaker and I think could have explored the dataset in a more interesting way.

- I found some results a bit confusing. Cross-Entropy and Focal losses were compared, but Cross-Entropy reportedly produced better results in several cases. Is there any insight into why this was the case given that the hypothesis is that focal loss should have been superior?

  Tables 4 and 5 mix results from various model architectures and the two loss functions - only the best top-k result for each dataset is reported. Please report results for all model variants at least in the supplementary information (architecture + loss). From inspecting the code in the repository, I assume this information is already available in order to select the best results in the table. I would also add a clarifying statement in the manuscript that the preliminary crop/non-crop experiments were performed using ResNet50 only (this is in the supplementary information already).

- I would like to see a confusion matrix (the code [exists](https://github.com/zahrag/BIOSCAN-1M/blob/cae9e6fb56bd803c78088b37ff52120468e145a6/visualize_results.py#L68)) at least for the order-level results. Accuracy is not a revealing metric here because the majority class (Diptera) is 80% of the dataset and unsurprisingly the model has perfect accuracy (to the precision reported in the paper). Clearly the model is not just predicting Diptera for all samples, but I'm curious to see if certain orders are confused. If it is straightforward, I think it would be beneficial to report F1 or a similar metric in addition to aggregated accuracy. Again, as no models are published (that I could easily find), I was unable to verify this myself.

- Note: I assume that whether cropping is helpful or not is largely dependent on what portion of the image is taken up by the subject. If the region of interest is small (relative) to the full 4K microscope image, then one would expect cropping to make a big difference. However for larger samples that fill the microplate well, it should matter less. I found it interesting that cropping only appears to provide a very minor improvement (1-2 percentage points at most), but it certainly saves space if most of the image is background.

  One speculation is that by cropping the images to a fixed size in pixels, you have effectively removed any scale information present in the image (as now the pixel scale varies significantly between samples). Presumably the raw images are all taken from a fixed focal length and it might be possible for a model to learn how big a specimen is with respect to the size of the well. Though while this might give you better results on the internal validation set, it would only be valid for images taken using a particular instrumental setup unless scale was explicitly passed to the model somehow.

- Minor note: random crops were used during training, but a 224 px crop of a 256 max-side image doesn't allow a huge degree of variation. The training transforms used were random H-flip and crop. You might see a boost in accuracy if rotations and other flip/mirror operations were included.

- In line 63/64 the wording in:

  > we designed and implemented a deep model, classifying BIOSCAN images into their taxonomic ranking, to serve as a baseline for future work utilizing this dataset.

  is misleading to me. The authors train models that can classify images into family and order, but I don't think it's fair to say "taxonomic ranking" as in the case of BIOSCAN-Insect, only the order is predicted, and in the case of BIOSCAN-Diptera, the order is fixed by definition. I think this claim would be correct if a model was presented that predicted both order and family over the entire dataset (e.g. family-level but including non-Diptera examples), but as-is, this statement should be reworded.

**Documentation:**

The work is well-documented, subject to some clarification questions. I had no issues obtaining the dataset or code and appropriate platforms are used to host each. I did not try to reproduce the results, but training scripts are provided.

**Ethics:**

With respect to my previous question about label provenance. If human annotators/taxonomists labelled the dataset, were they compensated and if so, fairly? How many annotators were employed to label 1M images? What was their labeling strategy (e.g. was assistive technology used like ML-in-the-loop, comparison to the DNA barcode).

**Limitations:**

1. Most samples in the dataset are classified only to the family level and it is unclear from the paper why finer-grained classes are not provided if the data are available as this is another step that researchers will have to take (e.g. querying BOLD).

2. Samples were only acquired from three countries. It would be interesting to see some discussion about use-cases for this dataset in the context of other large out-of-domain biological datasets (e.g. iNaturalist). For example, how well would a classifier trained on this dataset work on non-microscopy images? Presumably existing datasets/baseline models such as iNaturalist also contain some class overlap: would we expect a model trained on other insect imagery to also perform well at the order level?

3. The authors highlight that biological datasets offer an opportunity to test hierarchical models/losses, but that it's out of the scope of the paper. This is a shame, because I think it would be a lot more interesting than an off-the-shelf classifier. Even a non-hierarchical classifier trained to predict `Name` (e.g. each class is `order_family` with 10952 classes) would be worth exploring in my opinion.

4. The paper states

> ... also provides the Subfamily and Subspecies rank

however in the supplementary information, Table S2, 862,831 images do not have a subfamily and 1,128,300 images do not have a subspecies. The manuscript says there are 1.28M(?) images in the dataset, so it's incorrect to say that subspecies is provided and in 2/3 cases, subfamily is not provided either. Please update this wording to accurately reflect this.

5. An ethics statement is provided in supplementary information, but no societal impacts are discussed. One potential societal impact of automated barcoding techniques is how relevant human taxonomists are in a world of low-cost, fast sequencing. As I mentioned elsewhere, one assumes that DNA sequencing provides a higher confidence than a human annotator, but presumably there are cases when humans are currently able to provide insight/confirmation of results. Note this is more of a provocation for discussion than criticism.

**Opportunities For Improvement:**

I have split more specific issues into other subsections.

1. Are the model weight checkpoints released for the classifiers and the DETR crop model, and if not is there a specific reason?

2. I found it unclear in the work how (exactly) the data were labelled. The accompanying website says in the title:

  > hand-labelled insect images

  while the paper says

  > The taxonomic classifications (labels) of the dataset are created by matching the generated barcode to a reference library on the Barcode of Life Data System (BOLD) at the Centre for Biodiversity Genomics in Canada.

  > As an example, the Family group of the BIOSCAN-1M Insect dataset is indexed by 494 distinct families, however, there are 16,067 data samples that are not associated to any of these families, since they were not classified by human taxonomists.

  and the supplementary work says

  > The BIOSCAN project started by the International Barcode of Life (iBOL) Consortium, has collected a large dataset of hand-labelled images of insects. Each record is taxonomically classified by human experts, and accompanied by genetic information

  Were all 1 million images annotated by taxonomists/entomologists? By barcode matching? Combination of the two? I would assume that barcoding is the gold standard either way, but if the authors could make this clearer it would be helpful. In line 186 it is stated that all samples are sequenced which is presumably required to generate a BOLD identifier. If sequencing was deliberately not used to assign a label, it would be useful to explain why not. Simply reporting the difference between human/barcode classifications would be an interesting result on its own (see for example L76-81 where the authors discuss shortcomings of human labellers).

  I picked a random example from the top of the label TSV (`sampleid: BIOUG66193-B10`, `processid: GMMVA9932-21`, `BOLD:AAG8583`) which was annotated as Insecta Lepidoptera, but this barcode is associated with a [known species](https://v3.boldsystems.org/index.php/Public_BarcodeCluster?clusterguid=BOLD:AAG8583). Please provide some more information on why finer-grained classification at the genus/species level is not provided when it is apparently available for some samples.




**Relation To Prior Work:**

There are relatively few biological datasets of this kind, though many are also large and it is straightforward to "generate" arbitrary datasets using tools like GBIF/iNaturalist beyond the "official" competitions. BIOSCAN-1M contains novel imagery and I think is a good complement to existing publicly available work, especially with DNA sequence information which is not trivial to obtain. It would have been nice to see a comparison of any class overlaps between existing insect datasets.

iNaturalist's 2018 repository actually reports [8142](https://github.com/visipedia/inat_comp/blob/master/2018/README.md) classes. I believe the 5k reported in this work refers to [iNaturalist 2017](https://arxiv.org/pdf/1707.06642.pdf), which in any case is 5089 and not 5000.

- Please include/cite the more recent iNat 2018 competition.

My main concern is that iNat17/18 reports genus-level classes (species level was hidden for the competition I believe), while in Table 1 we compare against BIOSCAN at the order and family levels. This repeats for other datasets - Pl@ntNet reported 1081 _species_, but the number of order/families is a different statistic.  While there is a footnote for BIOSCAN, it doesn't make the numbers comparable and I think non-experts who are unfamiliar with biological taxonomy would definitely find this confusing.

One approach would be to summarize statistics by rank, but at least seeing number of orders / families for each dataset might make more sense, since BIOSCAN does not currently provide species. Frankly I'm unsure what the best/most informative route is, but I think the table should be revised for consistency regardless.

- Please report more consistent and comparable statistics between datasets.

**Summary And Contributions:**

The authors present a ~1M image dataset of insect microscopy imagery with associated taxonomic labels and genetic barcodes. Sample classifier training code is provided on Github (publicly) and an arXiv preprint is available (i.e. this was reviewed non-blind). The paper is well-written and easy to read, and I was able to easily access the data via the provided Zenodo links. As far as I can tell, model checkpoints are not published.

The reported evaluation is OK, but it's a minimal baseline and a missed opportunity for some potentially more interesting results. I have also reduced my score slightly because some of the paper's claims are misleading with respect to dataset content and should be revised for clarity - primarily that upon first reading, one might assume that species level information is provided for all/most samples when in fact only family/order labels are given. See criticisms and suggestions in the sections below which I think ought to be addressed before acceptance.

I think this is a good candidate for inclusion in the track, but I have a few clarifying questions. I would rate my evaluation confidence at between 4-5. My rating is primarily based on the potential utility of the dataset and would be nudged up 1-2 points were the issues I've raised fixed.

---

> ### Author Response · Authors · 2023-08-20
> **Rebuttal by Authors-Part1**
>
> We highly value the reviewer's comprehensive assessment and detailed feedback. Please find below our responses addressing the comments and questions provided by the reviewer:
>
>
> - We thank the reviewer for their useful suggestion. The checkpoints for the classifier, and DETR crop model are added to the GoogleDrive folder under a directory named BIOSCAN_1M_Insect_checkpoints.
>
> - Regarding dataset labelling procedure, the Order level classification is conducted by taxonomists and entomologists primarily relying on morphology rather than barcode matching. For Family and lower classifications, a combination of approaches may be employed, often supported by BIN assignment. In response to the reviewer's example, the case with random assignment lacks a definitive classification. The webpage referenced demonstrates that ambiguity emerges even at the Family assignment level. Consequently, this particular case was not subjected to further classification. Please see our revision added to the Supplementary material, section S6.6.1.
>
> - Several factors contribute to why most samples of the dataset are classified only to the family-level and finer-grained classes are not provided, with the primary one being the time constraints associated with completing the assignment. The complexity of the task arises from the fact that relying solely on a barcode is often insufficient due to potential ambiguities, as discussed earlier. Each sample's labeling requires verification through visual inspection (images), or in some cases, examination of the original specimen, before proceeding with further classification. This process is not easily scalable, prompting the adoption of BINs as a Species proxy. Please see our revision added to the Supplementary material, section S6.6.1.
>
> - The BIOSCAN project is currently in its initial stages, with its primary goal being the facilitation of a global biodiversity assessment. Through the utilization of the preprocessing module introduced in this paper, the cropping tool, and employing machine learning techniques for domain adaptation/generalization, one can leverage the pre-trained model on BIOSCAN_1M_Insect images to address out-of-distribution classification challenges. Please see our revision added to the Supplementary material, section S6.6.2.
>
> - We acknowledge the presence of various research ideas that could be explored using the BIOSCAN_1M_Insect dataset, aligning with our future plans. However, in the context of the current article, our primary focus is to release the dataset and showcase its inherent potential. To accomplish this, we include baseline experiments that illustrate its application with advanced machine learning models. Noted that "Name" is not considered as a defined taxonomic ranking group as it hosts the name of the lowest known rank entry, so for instance, if there exists a Species name of an organism, "Name" will show that, but if the organism only has Family level ID, then "Name" will show the ID. Please see our revision added to the Supplementary material, Section S6.6.2.
>
> - The reviewer's observation is accurate, and indeed, this limitation exists in the current form of the BIOSCAN_1M_Insect dataset, which is presented in the revised manuscript, Table 1, and discussed in the revised paper. It's important to note that we have outlined the taxonomic ranking groups, which will be progressively employed to label images within the dataset over time.
>
> - Regarding the automated barcoding techniques, taxonomic experts (human professionals) are engaged in this process. Their level of involvement tends to increase when the placement of a specimen is more contentious. Additionally, these experts are responsible for describing species, a task not handled by the machine. They also supply the reference ID that enables us to establish matches. Please see our revision added to the Supplementary material, Section S6.6.1.
>
> - We appreciate that the reviewer understands the effort that went into collection and curation of the dataset and preparation for its use by the machine learning and computer vision communities. We agree that the baseline problems and methods we explored were chosen to be simple and accessible and as a result, limited. They are not the focus of the paper and we expect future works will use the dataset for interesting problems such as hierarchical classification, zero-shot classification, set-valued classification and methods that improve performance in the fine-grained and long-tailed label regime. Please see our response to Reviewer 3 regarding data provenance. Please see the revision added to the Supplementary material Section S6.6.

---

> > ### Author Response · Authors · 2023-08-20
> > **Rebuttal by Authors-Part2**
> >
> > Continuing from the previous correspondence, we have outlined our responses addressing the reviewers' questions and concerns below:
> >
> > - Our results are presented based on the performance of the top-performing model, determined through the validation set. Our experiments encompassed a total of 24 trials, for the classification tasks of Insect-Order and Diptera-Family, utilizing three dataset variations: Large, Medium, and Small. We designed 4 unique models, using  2 loss functions (Cross-Entropy and Focal) and 2 distinct pretrained backbones (ResNet-50 and ViT-Base-Patch16-224).  We have added our full experimental results to the Supplementary materials (see Section S6.3). We have also updated our experiments to include runs with multiple  seeds.  We report the average and standard error across three runs for each condition. For the main paper, we select the model with the highest average performance on the validation set and report the results on the test set. Overall, we find that the ViT backbone with CE outperforms the other variations.
> >
> > - For the CE vs Focal loss, we did not see any improvement from using the Focal loss, and the results with the CE is actually a bit better than with the Focal loss. This could be because we did not carefully tune the hyperparameters (alpha and gamma) for the Focal loss. It’s also possible that a better way to address the class imbalance problem would be to sample the rarer classes during training since that will allow for infrequent classes to be seen more often during training.  With Focal loss, it’s possible that the rarer classes are seen so infrequently that the re-weighing is not having a positive effect. We elaborated on this aspect in the Supplementary material Section S6.3.
> >
> > - We have made clarification in the revised manuscript (Section 4.2) about using ResNet50 to conduct crop/non-crop experiments.
> >
> > - Thank you for the useful suggestion. We added the confusion matrices of our test performance to the Supplementary material Section S6.4. We have also reported the F1 scores in the revised Supplementary material Section S6.3 for the validation set and also in the revised manuscript Table 4 for the test set.
> >
> > - We appreciate your thoughtful insights. You're correct in pointing out that, within a fixed focal length setup, cropping could potentially lead to a loss of size information. To enhance the cropping tool's functionality, we are actively working on incorporating scale information associated with the crop. This involves utilizing the crop bounding box and an estimated pixel-to-metric scale specific to the given setup. While this enhancement is currently in progress and not reflected in our current experiments, its implementation aims to address the size normalization concern. Indeed, in scenarios where a specific metric scale per pixel isn't available, cropping does contribute to normalizing image sizes. As you rightly highlighted, while the performance improvement from cropping is somewhat modest, one significant advantage of the cropping tool is its ability to substantially reduce image sizes. This reduction in image size not only leads to a reduction in the time and computational resources required for experimentation but is particularly valuable when dealing with extensive datasets such as the Large dataset, which comprises over 1 million images. Given the combined benefits of improved performance and computational efficiency, we have conducted our primary experiments using the cropped images. This strategic choice aligns with the goal of optimizing both outcomes and resource utilization.
> >
> > - Thank you for bringing this to our attention; your perspective is indeed well-reasoned. We concur that integrating random vertical filtering and random rotation as part of data augmentation could potentially yield enhanced performance. Our decision to adopt the 224x224 random cropping and random horizontal flipping approach aligns with established practices in image classification tasks that involve ResNet models. Notably, insect images employing a pin for fixation typically avoid downward-pointing heads. In contrast, images taken against a petri dish background frequently feature such orientations. In these cases, integrating random vertical flips and random rotations for data augmentation seems pertinent. We intend to conduct experiments to assess whether these augmentations yield additional benefits. It's important to acknowledge that, while these augmentation strategies might not yield as significant improvements in the context of our current experiments with micro-averaged accuracy, this could be attributed to the already high performance achieved by our models. The exploration of additional augmentation techniques remains a valuable avenue for optimizing model performance.

---

> > > ### Author Response · Authors · 2023-08-20
> > > **Rebuttal by Authors-Part3**
> > >
> > > Continuing from the previous correspondence, we have outlined our responses addressing the reviewers' questions and concerns below:
> > >
> > >
> > > - We agree with the reviewer's suggestion and recognize the need for greater precision in our statement. We will revise it to accurately convey that "we devised and implemented a deep model for classifying BIOSCAN_1M_Insect images into specific taxonomic ranking groups...".  Thank you for pointing this out and the manuscript is revised accordingly.
> > >
> > > - The reviewer is correct that we were referring to the 2017 iNaturalist dataset, and that the more recent 2018 version includes 8142 unique categories. We have updated our table to refer to the most recent, 2021, version of iNaturalist, which features 10,000 species. Thank you to the reviewer for suggesting we analyse the amount of overlap between the insects in the BIOSCAN and iNaturalist datasets. Surprisingly, we found that there are only 153 genera and 62 species in common between the two datasets! This suggests the two datasets are highly complementary, and there is little overlap in the insects they feature. We have added this comparison to the paper. We note that only 7.5% of the BIOSCAN images are labelled at the species-level, and the BIN barcode data suggests there may be ten times more species present in the data than the number of unique species names we currently report. Thus we anticipate there are many times more insect species within the BIOSCAN dataset than iNaturalist. Please see the revisions in the paper Table 2 and under Section 2.4: Biological Datasets.
> > >
> > > - Thank you for the suggestion to improve the comparison with the existing datasets. We have revised the paper to show the breakdown of the number of unique categories found at each taxonomic rank within the BIOSCAN-1M dataset in a new table (Table 1 in the revised manuscript). We have added the name of the level taxonomic rank used for the categories/classes indicated in the pre-existing datasets described in revised Table 2 (formerly Table 1 in original version) which should enable the comparison to the number of unique values and samples at that level of granularity in BIOSCAN. Additionally, we have added the number of unique BINs in the BIOSCAN dataset as a point of comparison. This is salient because the BIN identity can be used as a proxy for the species identity. We have also incorporated the imbalance ratio as a point of comparison between datasets.
> > >
> > > - The annotators responsible for labelling the data were personnel affiliated with the Centre for Biodiversity Genomics, including technicians and research scientists, all engaged in full-time roles at the institute and receiving equitable compensation. A diverse team of approximately 15 to 20 individuals participated in providing labels for the dataset. These labels were derived from the images, forming the foundation for establishing higher-level taxonomies such as order and family. Additionally, the annotators conducted visual examinations of specimens for lower taxonomic ranks, utilizing primary literature as a reference where feasible. Please see our revisions in the Supplementary material S6.6.1.

---

### Official Review · Reviewer_qQMp · 2023-07-27
**Useful, novel dataset**

**Rating:** 7
**Confidence:** 5
**Clarity:** The paper is clear and easy to read.

**Strengths:**

This is a huge dataset of 1M insects. It's a very large amount of work to have collected and labeled this, and it offers significant opportunities for algorithmic innovation both in the tasks the authors present and in other potential tasks. The data is innovative here - e.g. iNaturalist is mostly not insects, and those insects that are there are photographed in the wild by amateurs, not as they would be collected by professional entomologists, and also there is no DNA data provided.

The baselines conducted are appropriate and perform well, though show opportunities for improvement, especially in the Macro accuracy.

**Additional Feedback:**

-

**Correctness:**

The dataset is thoughtfully constructed from both the ML and ecology perspectives.

**Documentation:**

Yes, detail is sufficient.

**Ethics:**

No ethical concerns.

**Limitations:**

I do not see limitations that should have been discussed.

**Opportunities For Improvement:**

Opportunities for improvement are overall fairly minor.

Results are currently presented only for the Vision Transformer architecture. These should be presented for the ResNet architecture as well.

Confusion matrices for each task would be helpful, perhaps in the supplementary material.

Diptera Medium seems to be significantly harder than Diptera Small (and Diptera Large). Why do you think that it?

Currently, it seems like the DNA barcodes are not being used directly in either of the tasks. (Though they were used for the identification of species.) Are there tasks that the authors envision will use these barcodes? This should at least be discussed.

**Relation To Prior Work:**

Prior work is adequately discussed.

**Summary And Contributions:**

The paper introduces a dataset of images of insects (primarily Diptera = flies), paired with genetic barcodes and family-level identifications. The authors conduct baseline experiments with ResNet and Transformer-based architectures on tasks involving classification to order/family level from images. Overall, this is a valuable and impressive contribution, representing a significant amount of work, tailored to be useful to both the ML and ecology audiences.

---

> ### Author Response · Authors · 2023-08-20
> **Rebuttal by Authors**
>
> The authors appreciate the reviewer’s evaluation as well as their useful comments. Below we address the concerns and questions.
>
> (1) Our initial results were presented based on the performance of the top-performing model, determined through the validation set. Our experimentation encompassed a total of 24 trials. These experiments were conducted for the classification tasks of Insect-Order and Diptera-Family, utilizing three dataset variations: Large, Medium, and Small. We designed 4 unique models, integrating 2 loss functions (Cross-Entropy and Focal) and 2 distinct pretrained backbones (ResNet-50 and ViT-Base-Patch16-224). The best performing models out of 4 were used at inference running experiments with our test data.
>
> Additionally, in response to the recommendation by reviewer 4KRL, and considering our time and resource limitations, in the revised version of the paper, we presented our 24 experiments results conducted using three distinct seeds. We computed the average performance across these diverse seed values. The model (ViT-Base-Patch16-224 with CE loss) exhibiting the highest average performance on the validation set was selected to undergo additional testing at inference and the test experiments' results are meticulously outlined in the revised paper Table 4. Moreover, the details of the validation experiments and results are presented in the revised Supplementary material Section S6.3.
>
> (2) Thank you for the useful suggestion. We have added confusion matrices of the model’s test performance to the Supplementary Material Please see Section S6.4.
>
> (3) As we maintained a uniform data distribution across datasets of various sizes and adhered to a consistent split mechanism, we fully agreed that the observed decline in the performance of Medium-Diptera-Family classification lacks justification, which could be due to a flaw in our experimentation and parameter settings. Consequently, we conducted a reiteration of these experiments, leading us to recognize that the outcomes initially presented in the submitted paper necessitate revision. Thank you for pointing this out and please find the updated results in the manuscript Table 4 and Supplementary material Section S6.3.
>
> (4) Yes. We plan using DNA barcode sequence and Barcode Index Number (BIN) in our future works. We have added a discussion of this future work to the Supplementary material - thank you for pointing out that this was not previously described in sufficient detail. Please find our revisions to the Supplementary material Section S6.6.

---

> > ### Comment · Reviewer_qQMp · 2023-08-29
> > **Thank you for the responses**
> >
> > Thank you to the authors for these responses. I continue to believe this paper represents a strong contribution and should be accepted.

---

### Author Response · Authors · 2023-08-20
**Rebuttal by Authors**

We extend our gratitude to all the reviewers for their thorough and constructive feedback. In response to their valuable input, we have meticulously revised both the manuscript and supplementary material. Notably, the sections where revisions have been made are indicated using a distinctive blue highlight.

---

### Decision · Program_Chairs · 2023-09-22

**Decision:**

Accept (Poster)

**Comment:**

All of the reviewers recommend accepting this paper. Authors highlighted that the dataset is very large and forms an important bridge between the machine learning and ecology communities. Other positive feedback was given to the public availability of the training code, appropriate baselines in the paper, and clear writing. Some minor issues and questions were raised (lack of species-level labels, generalization to other regions, annotation process, hierarchical losses, etc.) but most were addressed by the authors during the rebuttal. I recommend accepting this paper.